**Knowledgebase and Database Resources**

# *Cel*Est: a unified gene regulatory network for estimating transcription factor activities in *C. elegans*

Marcos Francisco Perez 🆔 *

Instituto de Biología Molecular de Barcelona (IBMB), CSIC, Parc Científic de Barcelona, C. Baldiri Reixac, 4-8, 08028 Barcelona, Spain

*Corresponding author: Instituto de Biología Molecular de Barcelona (IBMB), CSIC, Parc Científic de Barcelona, C. Baldiri Reixac, 4-8, 08028 Barcelona, Spain. Email: mpbbmc@ibmb.csic.es

Transcription factors (TFs) play a pivotal role in orchestrating critical intricate patterns of gene regulation. Although gene expression is complex, differential expression of hundreds of genes is often due to regulation by just a handful of TFs. Despite extensive efforts to elucidate TF-target regulatory relationships in *Caenorhabditis elegans*, existing experimental datasets cover distinct subsets of TFs and leave data integration challenging. Here, I introduce *Cel*Est, a unified gene regulatory network designed to estimate the activity of 487 distinct *C. elegans* TFs—~58% of the total—from gene expression data. To integrate data from ChIP-seq, DNA-binding motifs, and eY1H screens, optimal processing of each data type was benchmarked against a set of TF perturbation RNA-seq experiments. Moreover, I showcase how leveraging TF motif conservation in target promoters across genomes of related species can distinguish highly informative interactions, a strategy which can be applied to many model organisms. Integrated analyses of data from commonly studied conditions including heat shock, bacterial infection, and sex differences validates *Cel*Est's performance and highlights overlooked TFs that likely play major roles in coordinating the transcriptional response to these conditions. *Cel*Est can infer TF activity on a standard laptop computer within minutes. Furthermore, an *R Shiny* app with a step-by-step guide is provided for the community to perform rapid analysis with minimal coding required. I anticipate that widespread adoption of *Cel*EsT will significantly enhance the interpretive power of transcriptomic experiments, both present and retrospective, thereby advancing our understanding of gene regulation in *C. elegans* and beyond.

Keywords: transcriptomics; TF activity; *C. elegans*; RNA-seq; motif conservation; comparative genomics; orthology

## Introduction

Transcription factors (TFs) have a central role in coordinating development, the maintenance of cell identity and in transcriptional responses to physical, chemical, or xenobiotic insults. TFs act by binding to specific sites in the genome, recruiting regulatory proteins, transcriptional machinery, or chromatin modifiers to influence the expression of their nearby target genes (Lambert *et al.* 2018). While in humans the targets of TFs may be distantly located along the chromosome due to phenomena including chromatin looping (Palstra and Grosveld 2012), the popular model organism *Caenorhabditis elegans* exhibits a simpler regulatory model whereby most regulation occurs by TFs binding directly to the promoter region of target genes proximal to the transcription start site (TSS) (Reinke *et al.* 2013). Indeed, previous analysis of the transcriptional effects of TF perturbations in the *C. elegans* intestine found that physical interaction between TFs and promoters often underlies direct control of target gene expression (MacNeil *et al.* 2015).

Due to their essential roles, TFs are strongly conserved across long evolutionary timescales. This has allowed model organisms like *C. elegans* to play a key role in the discovery and characterization of the effector TFs downstream of many pathways that play roles in mammalian development. For example, the Smad TFs that coordinate transcription downstream of the TGF-β signaling pathway were first described and named for their mutant phenotypes in *C. elegans* and *Drosophila melanogaster* (Padgett *et al.* 1998).

Interestingly, the preferred DNA sequence motifs of TFs are also often strongly conserved even when overall protein sequence similarity is low (Lambert *et al.* 2019). For example, after >500 million years of divergence, the human and *C. elegans* orthologues of the TF TFEB/HLH-30 are just ~44% similar but their core DNA binding motif has remained essentially constant.

A host of post-transcriptional mechanisms exist to modify TFs in ways that strongly influence their gene regulatory activity (Reinke *et al.* 2013). As such, the expression of TFs (at either the mRNA or protein level) often does not reflect their activity. For example, the FOXO orthologue DAF-16 is phosphorylated downstream of insulin signaling, which leads to its exclusion from the nucleus and consequent inhibition of regulatory activity without being differentially expressed (Cahill *et al.* 2001; Lin *et al.* 2001). Post-translational control of TF activity is particularly critical when physical, chemical or xenobiotic insults necessitate a rapid transcriptional response, such as in the case of the activation of HSF-1 by heat shock (Vihervaara and Sistonen 2014). However, although changes in TF activity can sometimes be inferred by visualizing changes in subcellular localization or by profiling genomic binding, they are only directly detectable at the transcriptional level by the concerted effect of altered TF activity on their various target genes.

Several existing methods can provide quantitative estimates of TF activity from gene expression data based on the expression of their target genes (Badia-i-Mompel *et al.* 2022). However, these

approaches also require prior knowledge of which genes are regulated by each particular TF (also referred to as a TF's regulon) in the form of a gene regulatory network (GRN; Garcia-Alonso *et al.* 2019). While great effort has been expended defining TF regulons in *C. elegans* via different experimental methods (Narasimhan *et al.* 2015; Fuxman Bass *et al.* 2016; Kudron *et al.* 2018, 2024), each individual dataset covers a distinct subset of TFs and integration of the disparate data available has been challenging. As a result, the *C. elegans* community has to date lacked a unified TF-target prior knowledge resource to enable TF activity estimation and unlock the immense mechanistic insights it can provide.

The aim of this study is to integrate the distinct genome-wide resources on physical TF-target interactions available into a unified *C. elegans* GRN, which could be used with existing methods for quantitative TF activity estimation. A benchmarking dataset of TF perturbation RNA-seq experiments was compiled with which to quantitatively assess the performance of different data sources, data processing strategies and methods for TF activity estimation. Applied to a validation dataset of RNA-seq experiments from various genetic, environmental, or physiological conditions, the resulting GRN, christened *Cel*EsT, recapitulated known TF biology and also highlighted the potential importance of previously overlooked TFs. TF activity estimation with *Cel*EsT will provide *C. elegans* researchers with powerful mechanistic insights from transcriptomic experiments via rapid analysis on local computers using R, *Python* or a dedicated *R Shiny* app made available to the community at https://github.com/IBMB-MFP/CelEsT-app. A step-by-step guide to using the *Cel*EsT app can be found on *protocols.io* (Perez 2024a).

## Methods

### Identities and total number of *C. elegans* TFs

Despite multiple published estimates (Fuxman Bass *et al.* 2016; Ma *et al.* 2021; Kudron *et al.* 2024), the quoted number of *C. elegans* TFs at 833 is derived from the curated list in the latest manuscript from the modERN consortium (Kudron *et al.* 2024). TF family annotations were derived from the wTF3 resource (Fuxman Bass *et al.* 2016).

### Assembly of TF perturbation benchmarking RNA-seq dataset

To compile a set of TF perturbation RNA-seq experiments for benchmarking, I used NCBI's *eUtils* tool suite to download summaries of all *C. elegans* RNA-seq experiments in the Gene Expression Omnibus (GEO) (Barrett *et al.* 2012) accessed with the following query:

"Caenorhabditis elegans"[Organism] AND "expression profiling by high throughput sequencing"[DataSet Type].

The search was conducted on the 15/01/2024 and yielded 882 entries. I examined each entry for experiments where the expression of a TF present in any of the three principal datasets was perturbed (either knockdown or overexpression) and that had matched control samples. A few intensively studied TFs were over-represented (e.g. DAF-16); to ensure that the benchmarking set was not too heavily skewed toward a small number of TFs I arbitrarily limited the number of individual experiments for any single TF to a maximum of 4, choosing to retain experiments that represented high quality datasets (in terms of replicates and sequencing depth) and a maximal diversity of perturbations, conditions, and genetic backgrounds. The result was a set of 90 experiments for 45 unique TFs from 73 distinct studies

(Supplementary Table 2). Note that a few of these TFs were ultimately "lost" from the tested GRNs due to having very few targets remaining after, e.g. HOT region exclusion, such that the maximum size of benchmarking set actually used in the study was 87 experiments for 42 unique TFs.

For each study, I noted the identifiers for TF-perturbed and control samples and downloaded the raw sequencing data using NCBI's command-line package *SRA toolkit* (Edwards 2021). Reads were aligned with the *HISAT2* package (Kim *et al.* 2019) to the *C. elegans* genome (version WS288). Gene-level read counts were computed with "featureCounts" function (Liao *et al.* 2014) from the *Subread* package by cross-referencing to the WS288 canonical geneset "gtf" file downloaded from WormBase. Gene-level read counts were imported into *R* and differential expression (DE) analysis was performed with the *DESeq2* package (Love *et al.* 2014).

I controlled for potential mismatches of developmental speed between control and treatment samples using the *RAPToR* package (Bulteau and Francesconi 2022), following the procedure outlined in the *RAPToR* vignette. Briefly, sample ages were estimated using *RAPToR* against the references provided in the associated *wormRef* package; appropriate references were selected according to the annotated developmental stage on GEO or the relevant paper. I then interpolated counts from the appropriate reference for a range covering 1 h before and after the age range estimated to be covered by the test samples. I estimated gene dispersions using *DESeq2* using only the test samples and then created a *DESeq2* object combining these dispersions with the test counts and interpolated counts from the reference. I then performed the DE analysis by fitting a model including the effect of time modeled by a spline, treatment (TF perturbation vs control), and batch (test samples or interpolated reads). The DE statistics for contrasting TF perturbation vs control samples were extracted—these statistics exclude any DE due to developmental differences alone. Note, DE analysis for two experiments on dauer larvae were conducted without accounting for developmental differences due to the lack of an appropriate reference on which to stage the samples. The DE statistics for all genes (not only those significantly differentially expressed) within each experiment constituted the benchmark dataset used in the benchmarking pipeline. Prior to use in the benchmarking pipeline, I reversed the sign of DE statistics from experiments involving TF overexpression, such that the directions of all perturbations appeared equal.

### Additional data resources excluded from consideration

An additional potential source of information about TF-target interactions was curated user-supplied regulatory interactions from the worm community platform WormBase (Harris *et al.* 2020), but these interactions are very few relative to those supplied by other resources and often represent indirect relationships. As inclusion of these interactions did not improve the performance of GRNs in the TF activity estimation benchmarking pipeline (data not shown), I did not consider the WormBase interactions any further. I also attempted to infer regulatory relationships by TF-target coexpression using SJARACNe (Khatamian *et al.* 2019), using as input >600 *C. elegans* wild isolate RNA-seq samples (Zhang *et al.* 2022), but I found the resulting GRN had no predictive capacity (Supplementary Fig. 9). Likewise, I extracted TF mode of regulation (i.e. whether activator or repressor) where available from gene descriptions in WormBase and UniProt. However, this information did not consistently improve network performance and so was discarded. As such the resulting networks from this study are unsigned, i.e. they do not assign a

mode of regulation to each TF. Nonetheless, the TF modes of regulation compiled from WormBase and UniProt can be found in Supplementary Table 1.

## Assembly of GRNs from TF ChIP data

I downloaded the processed ChIP-seq TF-binding data for all experiments reported by the modERN consortium (Kudron *et al.* 2018, 2024) from epic.gs.washington.edu/modERNresource/. Six TFs are listed in the Supplementary Materials of Kudron *et al.* 2024 as having been successfully ChIPed but were not present in the Peaks file, presumably due to erroneous omission. Of these, 4 were present on the ENCODE platform (B0019.2, W05B10.2/ CCCH-3, T26A8.4, and W05H7.4/ZFP-3); I downloaded the processed "narrowPeak" files from the ENCODE website. Two additional TFs (W03F9.2 and Y116A8C.19) were present in the original modERN release (Kudron *et al.* 2018) (epic.gs. washington.edu/modERN) but were absent from the latest release, but without any sign that the data had been revoked or that they were no longer considered TFs. As such, they were included in this study and the data was obtained from the original modERN release.

Additionally, I downloaded summaries of all *C. elegans* ChIP-seq experiments from GEO (accessed 04/02/2024), a total of 604 experiments. I examined each entry for pulldowns of TFs and matched input controls. Of note, a number of early experiments from modENCODE/modERN were found to be present in GEO but these were ignored. In some cases, the data was already present in the modERN resource platform; in some cases it was not present but the ChIP for the TF was annotated as failed in the Supplementary Material of Kudron *et al.* (2024) or marked as revoked on the ENCODE platform. I identified ChIP data for an additional 6 TFs (F22A3.5/CEH-60, F56F11.3/KLF-1, T05C1.4/CAMT-1, T23D8.8/CFI-1, ZC376.7/ATFS-1, and Y51H4A.17/STA-1); however, the data for CFI-1 was ultimately excluded due to an unusually high number of binding targets. Raw sequencing data for TF ChIP and input controls were downloaded with *SRA toolkit* (Edwards 2021) and aligned to the *C. elegans* genome (WS288) with *bowtie2* (Langmead and Salzberg 2012). I then used the *Genrich* package to call peaks, as *Genrich* allows integrated peak calling against a control file using multiple replicates. The resulting "narrowPeak" files were then treated as for those downloaded from ENCODE.

Potential targets of TF ChIP peaks were assigned manually, ignoring the annotated targets present in some processed files from modERN. From the *TxDb.Celegans.UCSC.ce11.refGene* package/object in R (Bioconductor Core Team 2019), I obtained genomic locations for protein-coding genes. As the objective was to have potential targets with robust and differentiated expression in order to be useful as markers for inferring TF activity, I censored these potential genes by those whose expression was detected in at least 2/3 of the ~600 samples in the largest available *C. elegans* RNA-seq dataset triplicate samples from ~200 wild strains of the *C. elegans* Natural Diversity Resource (CeNDR; Zhang *et al.* 2022). I also eliminated those genes that were annotated as being downstream in a *C. elegans* operon in Allen *et al.* 2011, as they would not likely be controlled by proximal promoter regions. This resulted in a set of 14,083 potential target genes. I derived the genomic regions taken as corresponding to proximal promoter regions for these genes by taking 1,000 bp upstream and 200 bp downstream of the TSS using the "genes()" and "promoters()" functions of the *GenomicFeatures* package (Lawrence *et al.* 2013) in R to identify overlaps between ChIP peaks and these candidate target promoter regions. Of note, multiple interactions could derive from

a single ChIP peak if it fell in a region where two promoters overlapped; i.e. rather than assigning each individual peak to a single definitive target as in the processed modERN files, I counted all TFs bound to each promoter as potential regulators regardless of any greater proximity of a peak to the TSS of a different gene.

ChIP peaks for a given TF were prioritized into an ordered list based first on repeated occurrence across multiple independent experiments, if applicable, and then based on the strength of the ChIP peak obtained from the narrowPeak files. Cutoffs were then applied, taking the top targets corresponding to a given cutoff from this ordered list, to generate distinct GRNs for benchmarking. Any TFs with fewer than 15 targets were filtered out of the final GRN to prevent spurious TF activity estimations resulting from linear model fitting to few data points.

Genomic locations of high occupancy target (HOT) regions were taken as defined in Supplementary Table 10 of Kudron *et al.* (2018). Of note, this table listed the number of experiments for which a given region was bound by a TF, but repeated occurrence of binding of the same TF in multiple experiments were considered as independent instances counting toward this total and thus to the definition of a HOT region. As I did not consider that a region should be excluded in part due to consistent binding of those TFs which had multiple experiments, I instead counted the number of distinct TFs, of the total 217 assayed, annotated as binding the HOT region in any experiment. GRNs were generated excluding as potential targets any gene whose promoter overlapped a HOT region, defined by binding of a given cutoff of distinct TFs out of the total 217 assayed in that study. As shown in Supplementary Fig. 1d, the best performance (considering trade-offs with "loss" from the network of those TFs with few remaining targets) came from applying an exclusion criterion of 50/217 TFs to regions considered to be HOT regions.

## Assembly of GRNs from in vitro DNA-binding motifs

I downloaded all available DNA-binding motifs for *C. elegans* TFs from CisBP (Weirauch *et al.* 2014) (accessed 23/02/2023). I excluded any motifs derived from ChIP-seq data, retaining only those obtained via in vitro methods (principally protein-binding microarrays and systematic evolution of ligands by exponential enrichment experiments). Some motifs were directly measured from that TF's DNA-binding domain, while others were indirectly inferred from similar DNA-binding domains from other species. I excluded any indirect motifs that derived from directly-measured motifs of other *C. elegans* TFs to avoid motif duplication. Likewise, in the case where a single motif from another species was assigned to multiple *C. elegans* TFs, I manually selected one TF based on criteria of best similarity score in CisBP and expression level in *C. elegans* and excluded the other TFs.

Candidate target promoter regions were identified as described above. The sequences corresponding to these promoters were obtained from the *BSgenome.Celegans.UCSC.ce11* package/object using *GenomicRange's* "get_sequence()" function in *R*. I ran the FIMO tool from the *MEME Suite* collection of sequence motif tools (Bailey *et al.* 2015), searching for instances of the CisBP TF motifs in the promoter sequences. Promoters were ordered according to the best-scoring motif match returned by FIMO before cutoffs were applied to generate GRNs. For calculating homotypic binding scores taking into account additional matches beyond the first, points for additional matches were added to the score for the best match with exponentially diminishing returns for additional matches. The score of the second-best match was added divided by $5^1$, of the third-best match divided by $5^2$ and so on. The

diminishing returns parameter 5 was selected as the best-performing by benchmarking GRNs with scores computed with different diminishing returns parameters (not shown).

## Assembly of GRNs from eY1H data

eY1H data were taken from Supplementary Table 2 of Fuxman Bass *et al.* (2016). I filtered interactions to only those annotated as belonging to the "high-quality dataset." This gave 21,864 interactions for 366 unique TFs. After filtering out TFs with fewer than 15 targets, there remained 20,773 interactions for 160 TFs.

## Assembly of GRNs from SJARACNe-inferred coexpressed TF/gene pairs

Sample ages for ~600 RNA-seq samples from CeNDR wild *C. elegans* strains (Zhang *et al.* 2022) were inferred using *RAPToR* from TPM-normalized values downloaded as a Supplementary File from GEO entry GSE186719. Sample transcript-level raw counts (also obtained as a supplementary from the same GEO entry) were summed to generate gene-level counts, which were then processed with a variance-stabilizing transformation using the "vst()" function of *DESeq2* in *R*. The vst-normalized values were modeled as a function of time with the "smooth.spline()" function in *R* with 6 degrees of freedom to remove any differences between samples due to differential developmental age. The residuals were extracted from the model and used as input values for SJARACNe (Khatamian *et al.* 2019). Note, I also repeated this analysis with values with no correction for developmental age without any improvement in performance (not shown). I filtered all genes to remove any genes that were not detected in every sample. I supplied a list of nodes (TFs) to SJARACNe that consisted of all of the TFs present in any of the other three datasets, except those not detected in every sample (leading to exclusion of 123 TFs). I ran SJARACNe on the command line and filtered the output for interactions with a *P*-value below 0.0001. A GRN was then constructed with the remaining interactions for each TFs, weighted according to the sign of the interaction detected by SJARACNe.

## Assembly of combined GRNs from multiple data sources

When combining GRNs from different datasets, I always used unfiltered GRNs (i.e. that did not have TFs excluded due to low target numbers); this was because a TF might be able to reach the threshold for inclusion (15 targets) by combining targets from multiple data sources, although it had failed to reach this threshold so in any single data source. For unweighted GRNs target lists for each TF were simply combined (with removal of duplicate interactions); for weighted GRNs interactions were given weights according to their appearance in the three datasets. Interactions appearing in all three data sources were given a maximum weight of 1, while interactions appearing in 2 or 1 data sources were given weights of 0.67 and 0.33 respectively.

## Benchmarking

In order to benchmark the performance of the GRNs against our TF perturbation benchmarking set, I used the *decoupler* package implemented in *Python* (Badia-i-Mompel *et al.* 2022). Note that the benchmarking pipeline is more fully developed and much faster in the *Python* version of the *decoupler* package relative to the *R* version. I used the "benchmark()" function, using the methods "mlm," "wlm," and "wsum," also computing a multimethod "consensus" score. In order to compare the benchmarking output against randomly shuffled networks, I used the "shuffle_net()" function of *decoupler* to produce 100 randomly shuffled networks

for each of our tested GRNs. Benchmarking was then repeated for each of the shuffled networks. The plotted points and error bars for random networks represent the mean and standard deviation of the performance metrics (AUROC/AUPRC) for the 100 shuffled iterations of that network.

## Biases of target genes derived from ChIP-seq or promoter motifs

To investigate differences in chromatin domains in which target genes derived from different datasets were located, genomic coordinates from the *TxDb.Celegans.UCSC.ce11.refGene* package/object in *R* were cross-referenced with domain coordinates from Supplemental Data Table 2 from Evans *et al.* (2016) using the "findOverlaps()" function of the *GenomicRanges* package. When genes were annotated to active or regulated chromatin domains, enrichments were judged against the location of all unique target genes using a $\chi^2$ test with the "chisq.test()" function.

To calculate the cross-tissue Gini coefficient for target genes, we used the pseudobulk tissue-specific expression values compiled from single-cell RNA-seq data from Supplementary Table 3 of Cao *et al.* (2017) with the "Gini()" function of the *DescTools* package in *R*. Distributions for targets identified by ChIP-seq or promoter motifs were compared by a Kruskal–Wallis test with Dunn's post-hoc test against the distribution of Gini coefficients from all unique target genes to derive pairwise *P*-values.

To calculate enrichments of tissue-specific genes among the targets derived from ChIP-seq or promoter motifs, we used Supplementary Table 7 of Cao *et al.* (2017), counting genes as tissue-specific if they had a ratio of gene expression from the most-expressed cell type to the second-most-expressed cell type of >5 and a *q*-value < 0.05. Cell type annotations were combined into tissue-level annotations. Odds ratios and *P*-values were then determined by a Fisher's exact test using the "fisher.test()" function against all unique target genes.

## Cross-species conservation-based filtering of TF targets

In order to perform cross-species conservation-based filtering of potential TF targets, I first build *BSGenome* (Pagès 2024) and *TxDb* packages in *R* for each of the 10 additional *Caenorhabditis* species (*brenneri, briggsae, inopinata, latens, nigoni, remanei, sinica, tribulationis, tropicalis,* and *zanzibari*; genome versions in Supplementary Table 7) in order to obtain sequences corresponding to potential target promoters. The genome assemblies for these species were downloaded as FASTA files from WormBase ParaSite (Howe *et al.* 2017). Chromosome/contig numbers were extracted and the genome-sequence FASTA file for each species was split into individual FASTA files for each chromosome/contig. I then created a seed file and used *BSGenome*'s "forgeBSgenomeDataPkg()" function to create a "BSGenome" data package for each species. I also created a "TxDb" genome annotation object for each species using annotation files downloaded from WormBase ParaSite using *GenomicFeatures*' "makeTxDbFromGFF()" function.

I then obtained the genomic locations of the promoters of genes as described above for *C. elegans* using the "promoters()" and "genes()" function of *GenomicFeatures* with the corresponding species' "TxDb" annotation data object. I filtered out any promoters which were incomplete due to proximity to the edge of a chromosome/contig. For each species, I restricted the set of potential target promoters to those that were annotated as one-to-one orthologues of a *C. elegans* gene in Ensembl *BioMart* (Kinsella *et al.* 2011) queried using the *biomaRt* package in *R*. I also filtered out candidate promoters that were annotated as downstream in

an operon in either *C. elegans* as described above or *C. briggsae* (annotations for the latter obtained from Supplementary Table 7 of Jhaveri *et al.* (2022)), as operons show a considerable degree of conservation across *Caenorhabditis* species (Qian and Zhang 2008; Pettitt *et al.* 2014) and so genes contained downstream within operons in at least one *Caenorhabditis* species are less likely to be controlled by the sequences proximal to the annotated TSS. I then obtained the sequences for the set of filtered promoters using "get_sequences()" and the corresponding species "BSGenome" data object.

I then searched for all *C. elegans* TF DNA-binding motifs obtained from CisBP as above on the promoter sequences for each species using FIMO as described above. For each TF, I extracted the best motif score for each potential target promoter in each species. If the potential target promoter sequence was absent in a species (due to the absence of an annotated one-to-one orthologue or due to filtering out incomplete promoters), then it was marked with NA in that species; if a potential target was present but exhibited no motif match it was marked with a 0. If there was not a one-to-one orthologue of the TF itself in that species, then all potential targets were marked with NA for that TF in that species. I calculated a conservation score as follows. First, I transformed the best motif match score from FIMO for each target into a rank-quantile for that motif and that species (the promoter with the best matching motif score in a genome had a rank-quantile close to 0, whereas promoters with no motif match had a rank-quantile of 1). I then computed the product of these rank-quantiles for each TF-promoter combination across all species to obtain the conservation score for that target and that TF.

I calculated the probability of conservation of a motif's presence across species in orthologous targets as follows. For each TF, the rank-quantiles of promoters in each species were shuffled randomly between all non-NA targets (thus preserving the distribution of motif scores within each species and the number of species considered for each promoter). An empirical random distribution of conservation scores was generated by calculating the product of the rank-quantiles for 10,000 iterations of random shuffling. The observed conservation scores were compared to this random distribution to obtain an empirical *P*-value for each TF-target pair. These empirical *P*-values were adjusted for multiple comparisons using the Benjamini Hochberg/method to generate FDR values.

I generated multiple GRNs using different initial cutoffs of *C. elegans* scores and then filtering down to those interactions with orthology-derived conservation probabilities meeting different FDR cutoffs. These GRNs were benchmarked to find the optimal cutoffs for *C. elegans* score and conservation probability. As the conservation-based GRN with the best performance used a seemingly high FDR of 0.8, these FDRs are likely conservative.

### Ordering of ChIP targets by de novo motif conservation across orthologues

I applied the "STREME" algorithm (Bailey 2021) from *MEME Suite* by running the function "runStreme()" from the *R* package *memes* (Nystrom and McKay 2021). As input sequences, I used the 100 bp surrounding the center of ChIP peaks, which was calculated by adding the "peak" value (10th column of ENCODE narrowPeak files) to the "start" genomic coordinate for each peak. A minimum of 100 ChIP peaks were used to extract de novo motifs.

To judge the performance of "STREME" in defining de novo motifs, I used the "Tomtom" algorithm from *MEME Suite* to compare de novo motifs to directly determined motifs from CisBP. I

observed much better performance when excluding ChIP peaks located in HOT regions (cutoff of 50 applied as described above), as found elsewhere (data not shown; Gerstein *et al.* 2010; Kudron *et al.* 2024). I also experimented with using a maximum of the top 300 ChIP peaks, rather than all peaks, for the de novo motif search with "STREME". Restricting the ChIP peaks in this way led to fewer motifs discovered per TF (mean of 4.32 vs 8.08 with all peaks). Although the motif discovery rate (defined as the identification of at least one motif with a significant similarity to the known motif) was lower (27/77 TFs vs 31/77), the known motif was the top hit more often (14/77 TFs vs 12/77 TFs). As such, I restricted the ChIP peaks for de novo motif discovery to the top 300 non-HOT peaks.

I proceeded to calculate conservation probabilities as described above using the top de novo motif for each TF. Despite the top de novo motif matching, the known in vitro motif <20% of the time, ChIP GRNs with targets ordered according to conservation probability performed surprisingly well.

Additionally, I ordered ChIP targets by the conservation probabilities for the known motif, rather than the de novo motif, for the 119 TFs present in both the ChIP data and CisBP (either directly or indirectly determined motifs). Interestingly, these networks performed markedly worse than those using the top de novo motif.

### Application of *CelEsT* to RNA-seq datasets for genetic, environmental, or physiological conditions

I identified candidate datasets for insulin signaling mutants, heat shock, bacterial infection with pathogenic *Pseudomonas aeruginosa* strain PA14 and males vs hermaphrodites from the *C. elegans* RNA-seq datasets in the GEO. RNA-seq datasets were downloaded, aligned and gene-level read counts computed as described for the preparation of the benchmark dataset. Similarly, DE analysis accounting for developmental age was conducted as previously described. TF activity estimates were derived for each experiment using *CelEsT* with *decoupleR* in *R* with the experiment's DE stats as input and using the "mlm" method. I plotted TF activities as heatmaps and generated study correlation matrices of TF activities to detect and exclude any outlying samples (*daf-2(e1370)* mutants one study—GSE36041; *daf-16* null mutants two studies—GSE240821 and GSE108848; heat-shock one study—GSE122015; males vs hermaphrodites one study—GSE222447).

To amalgamate the results of the independent experiments for each condition, I converted the TF activities for each experiment into a z-score by subtracting from each TF activity score the mean of the scores for all TFs and dividing by the standard deviation of the scores. I then took the mean activity z-score for each TF across all experiments for a condition. Similarly, for each TF I took the geometric mean of the *P*-values computed in each experiment. Volcano plots show these mean activity z-scores against the $-\log_{10}$(geometric mean *P*-value).

### Plotting

Plots were rendered in *R* using the *ggplot2* (line/volcano/violin/scatter plots), *gplots* (heatmaps), and *eulerr* (Euler plots) packages.

## Results
### Establishment of GRNs based on large-scale datasets of direct TF-DNA interactions

To construct GRNs for *C. elegans*, 3 major resources were identified for selection of TF regulatory targets, originating from distinct

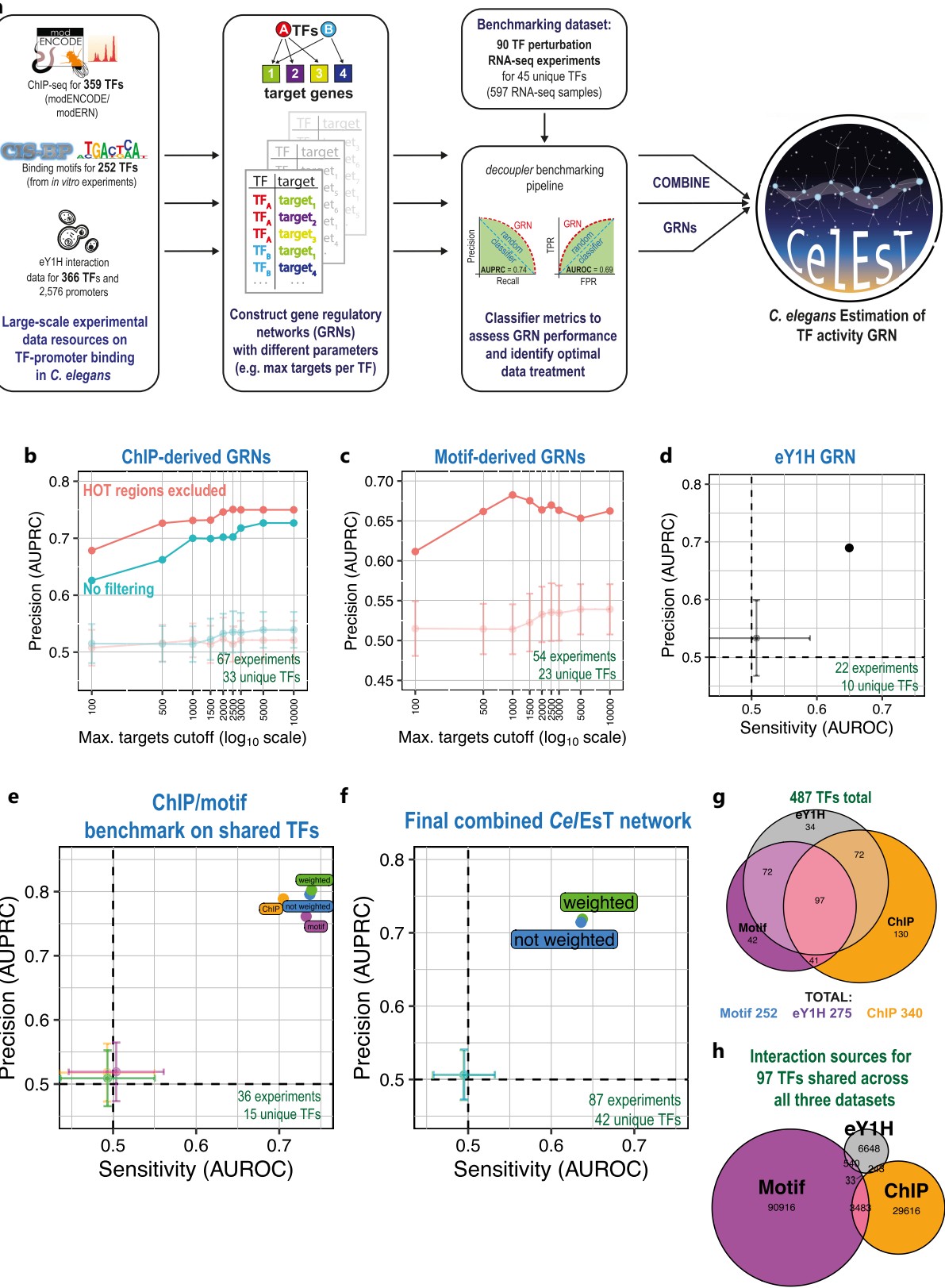

**Fig. 1.** Generation and benchmarking of GRNs from publicly available large-scale datasets. a) *Cel*EsT synthesises TF-target information from 3 distinct large-scale experimental sources: TF ChIP-seq experiments from the modERN consortium, TF DNA-binding motifs directly measured or inferred from in vitro experiments from the CisBP database and a dataset of enhanced yeast one-hybrid (eY1H) assays measuring direct TF-promoter interactions. GRNs are constructed and evaluated against a benchmarking pipeline using the *decoupler* software package. Optimized networks are combined to yield a unified network. b) Line shows change in AUPRC in benchmarking pipeline with increasing cutoff for maximum number of targets per TF for GRNs derived from ChIP-seq data with (upper line) or without (lower line) filtering of HOT regions. Note x-axis (cutoff) is on a log scale. Results for the multivariate

(continued)

experimental techniques (Fig. 1a). The first consisted of 525 ChIP-seq experiments for 354 TFs from the modERN consortium (Kudron *et al.* 2018, 2024); heir to modENCODE (Gerstein *et al.* 2010). The second was DNA binding motifs for 252 motifs identified by in vitro methods, either determined directly or inferred by high similarity of a TF's DNA-binding domain to other TFs with characterized binding sequence specificity (Lambert *et al.* 2019), from the CisBP database (Narasimhan *et al.* 2015). The third data resource was a collection of TF-gene promoter interactions identified from enhanced yeast one-hybrid assays (eY1H) (Fuxman Bass *et al.* 2016). In addition to ChIP-seq experiments from modERN, ChIP-seq datasets were identified for another 5 TFs not included in modERN from the NCBI GEO. In total information was recovered to identify potential targets for 596 of an estimated 833 *C. elegans* TFs (Kudron *et al.* 2024). Additional sources for TF-target interactions were considered but ultimately discarded (see Methods). The networks resulting from this study are unsigned i.e. they do not assign a mode of regulation (e.g. activator or repressor) to each TF. Nonetheless, TF modes of regulation compiled from WormBase and UniProt annotations can be found in Supplementary Table 1 and in the output of the *CelEsT Shiny* app.

In order to judge the performance of the GRNs that resulted from the integration of the three data resources described above, I used the benchmarking pipeline from the *decoupler* package (Fig. 1a; Badia-i-Mompel *et al.* 2022). This pipeline assesses the ability of TF activity estimations to correctly identify perturbed TFs based on the expression of their known target genes in a benchmarking transcriptomic dataset. This allows for a quantitative performance comparison both for different GRNs and for different statistical methods of estimating TF activity, using the classifier metrics AUROC and AUPRC. Although AUROC/AUPRC are related and correlated metrics, they can display opposite trends (e.g. Supplementary Fig. 1c) and so GRNs that maximized both AUROC and AUPRC were prioritized where possible. Two of the three statistical methods considered gauge TF activity by fitting a linear model to target gene expression. Of these, the univariate linear model ("ulm") method considers TFs separately, whereas the multivariate linear model ("mlm") method fits a single model to all TFs. The latter can thereby theoretically disentangle the separate effects of distinct TFs with overlapping targets. The weighted sum method ("wsum") involves summing scores for all targets of a given TF. The *decoupler* package also provides a consensus score, which integrates the scores from disparate methods. For more detail on specific methods, see Supplementary Table 1 of Badia-i-Mompel *et al.* (2022).

As a benchmarking resource, 90 *C. elegans* RNA-seq experiments were identified from the GEO repository with TF loss of function (e.g. by mutation or RNAi knockdown) or gain-of-function (via overexpression or gain-of-function alleles) with matching controls for 45 unique TFs from 73 distinct studies (Supplementary Table 2). RNA-seq datasets were uniformly

processed and DE analysis was performed between TF perturbation and control samples, including accounting for any potential mismatch in developmental stage resulting from the knockdown (Bulteau and Francesconi 2022; see Methods). The collected DE stats for all studies were used as the benchmarking dataset.

## Excluding targets in HOT regions improves performance of ChIP-derived GRNs

Potential TF targets from ChIP-seq experiments were identified by TF binding to the target promoter. Targets were prioritized firstly by recurrence over multiple experiments, if different life stages had been assayed for the same TF (~20% of modERN TFs), and secondly by strength of ChIP peak signal. Given that some ChIP peaks might represent false positives, the benchmarking pipeline was used to identify a suitable cutoff for maximum number of targets per TF (Supplementary Fig. 1a). However, I found that the best performance resulted from applying no cutoff to ChIP-derived regulons (Fig. 1b, Supplementary Fig. 1, b, c, e and f).

In large-scale ChIP-seq datasets, some genomic sites appear to be promiscuously bound by large numbers of TFs—so-called HOT regions (Kvon *et al.* 2012). Including targets found in HOT regions with many bound TFs could complicate the attribution of expression differences at these loci to activity of specific TFs. To address this, gene promoters within HOT regions were excluded (Supplementary Fig. 1d), substantially improving benchmarking pipeline performance at all cutoffs and for all TF activity estimation methods (Fig. 1b, Supplementary Fig. 1, b, c, e and f). However, this led to the loss of 44 TFs that retained little or no promoter binding outside of HOT regions.

Of note, comparison between TF activity estimation methods using ChIP-derived GRNs demonstrated that the multivariate linear model ("mlm") method had both the best overall performance and the best discrimination vs randomized networks, with substantially worse and near-identical performance for the univariate linear model ("ulm") and weighted sum ("wsum") methods (Supplementary Fig. 1b-f).

## Optimal regulon size cutoffs are critical for GRNs derived from TF DNA-binding motifs

To identify potential targets for 252 TFs with in vitro-derived DNA binding motifs, gene promoter sequences were scanned for significant motif sequence matches (Supplementary Fig. 2a). In contrast to ChIP-derived GRNs, network performance peaked at a maximum cutoff of 1000 targets for each TF (Fig. 1c, Supplementary Fig. 2b-c). While the "mlm" method was still the best performing individual statistical method for TF activity estimation, the motif-based GRNs also performed well with both "ulm" and "wsum" (Supplementary Fig. 2b-c), leading to even better performance from the "consensus" score, which integrates results from multiple methods.

Promoters may have several distinct binding sites for the same TF (Kazemian *et al.* 2013), which can lead to high TF binding

**Fig. 1.** (Continued)
linear model ("mlm") method are shown. Faded lines show AUPRC mean and standard deviation (SD) for randomized networks (100 iterations). See also Supplementary Fig. 1, b, c, e and f for performance with other statistical methods. c) Line shows evolution of AUPRC in benchmarking pipeline for the "mlm" method with increasing cutoff for maximum number of targets per TF (log scale) for GRNs derived from TF DNA binding motifs. Faded line shows AUPRC mean/SD for 100 randomized networks. See also Supplementary Fig. 2b and c for performance with other statistical methods. d) Benchmarking performance (AUPRC/AUROC) for a GRN with regulons derived from TF-promoter interactions demonstrated in an eY1H assay. Lighter point with error bars shows mean/SD for randomized network. See also Supplementary Fig. 3a for other methods. e) Benchmarking performance for ChIP- or motif-derived GRNs, or combined GRNs with or without extra weight for shared targets, benchmarked only on shared TFs to allow direct comparison. See also Supplementary Fig. 3b for other methods. f) Benchmarking performance for the *CelEsT* network, which covers 487 TFs by combining data from multiple experimental sources. See also Supplementary Fig. 3c for other methods. g) Euler diagram showing contribution of TFs to the *CelEsT* network from the three distinct experimental datasets. Total unique TFs shown above, total from each dataset shown below. h) Euler diagram showing data sources for TF-target interactions in *CelEsT* for 97 TFs present in all three datasets.

despite lower affinity of individual sites (Crocker *et al.* 2016; Shahein *et al.* 2022), a phenomenon called homotypic binding site clustering (Payne and Wagner 2015) (Supplementary Fig. 2d). I hypothesized that taking into account the presence of multiple binding sites for a given TF, rather than simply considering the best sequence match, might improve network performance. To reflect the possibility of homotypic binding, a combined score was computed providing additional but diminishing points for multiple significant TF motif matches in the candidate target gene's promoter region. In effect, this changes the ordering of potential target genes and so the composition of genes that are included within a given cutoff. While this combined homotypic binding score did improve performance using very low cutoffs, at higher cutoffs it had the opposite effect (Supplementary Fig. 2e). As such, I preceded using GRNs compiled from TF targets ordered using only the strength of the best sequence match to the interrogated motif.

Additionally, a GRN was compiled from 160 TFs with at least 15 target promoters from the eY1H TF-promoter interaction dataset in Fuxman Bass *et al.* 2016. This GRN displayed good performance in the benchmarking dataset, thus clearly contributing valuable regulatory interactions (Fig. 1d, Supplementary Fig. 3a).

## Integrated GRNs from multiple data sources have improved performance

The optimal ChIP- and motif-based GRNs both performed similarly well on a benchmarking set consisting only of perturbations of TFs common to both data sources. Combining the targets for the two GRNs led to improved performance (Fig. 1e, Supplementary Fig. 3b). Several TF activity estimation methods allow higher-confidence interactions in the GRN to be given extra weight. Weighting interactions by their presence in one or both data sources did indeed improve the performance of the combined GRN over simply combining all targets (Fig. 1e, Supplementary Fig. 3b).

The ChIP-, motif-, and eY1H-derived GRNs were integrated into a final network. Overall, this combined GRN covers 487 TFs, 58.5% of the total estimated *C. elegans* TFs (Kudron *et al.* 2024). The combined and weighted GRN performs well, with an AUPRC of 0.719 and an AUROC of 0.638 (Fig. 1f, Supplementary Fig. 3c), a performance comparable to popular human GRNs like DoRothEA (Müller-Dott *et al.* 2023). I call this final combined network *Cel*EsT (*C. elegans* Estimation of TF Activity; Supplementary Table 3). The *Cel*EsT network performed similarly well on a benchmarking set without any correction for developmental age (Supplementary Fig. 3d).

The ChIP, motif and eY1H datasets contributed similar numbers of TFs to *Cel*EsT (Fig. 1g), although there were more motif-derived interactions and few eY1H-derived interactions. Despite the fact that all data sources displayed good benchmarking performance in isolation, interactions were rarely shared even for those TFs that appeared in all three data sources (Fig. 1h), in agreement with previous findings (Garcia-Alonso *et al.* 2019). To investigate potential reasons for this lack of overlap, the genomic and expression characteristics of targets derived from different datasets were interrogated. ChIP-derived interactions were biased toward target genes that were located in active chromatin domains (Supplementary Fig. 3e; Evans *et al.* 2016) and that were expressed either broadly across tissues (Supplementary Fig. 3f) or in tissues that contribute disproportionately to whole-animal chromatin (up to 32n polyploid intestinal cells (McGhee 2007) and the multinucleate gonad/germline; Supplementary Fig. 3g). These biases were likely to be technical. Motif-derived targets

showed no bias for either chromatin context or expression across tissues (Supplementary Fig. 3e-g).

While most major TF families displayed good benchmarking performance, the zinc finger NHR and C2H2 families did not perform better than random (Supplementary Fig. 4a and b). I note however that the benchmarking set covers <2% of TFs in these families. Performances for individual TFs, while approximate due to tiny benchmarking sets, varied substantially (Supplementary Fig. 4c-d). Notably, performance of some TFs depended strongly on whether developmental age correction was applied. While the putative developmental clock gene BLMP-1 (Meeuse *et al.* 2020) performed markedly worse with age correction, which may remove some signal from BLMP-1-driven transcriptomic changes, ZTF-11 performed much better. The performance of the *Cel*EsT GRN was robust to removal of ~25% of TFs from the benchmarking set (Supplementary Fig. 4e and f) and thus was not largely driven by a small number of high-performing TFs.

## Conservation of TF binding motifs across species identified high-confidence TF targets

In the integrated *Cel*EsT network, each TF has an average of 829 targets (median 1,000) and each target in turn is regulated by an average of 27.5 TFs (median 23). For researchers interested in identifying fewer, more high-confidence TF targets, the benchmarking pipeline may offer an analytic resource to assess how TF-target relationships can be effectively distilled into a more highly informative subset.

TF binding motifs are strongly conserved across even distantly related species, despite apparent divergence at the protein similarity level (Nitta *et al.* 2015). By comparison, individual motifs in promoters are vulnerable to even a single point mutation and thereby undergo constant evolutionary turnover in terms of loss and gain (Tuğrul *et al.* 2015). However, biologically important TF-promoter interactions will be subject to selection that leads to the retention (or reappearance) of a specific motif in orthologous target genes across species (Villar *et al.* 2014). Strong matches to a TF motif within a promoter in one species may therefore be spurious but target promoters where the TF motif is conserved across target orthologues in multiple species likely represent important interactions with selectable fitness consequences. Such an alignment-free conservation-based approach to TF target identification has been previously employed in *C. elegans* (Glenwinkel *et al.* 2014).

Orthologous promoter sequences of *C. elegans* genes from 10 additional *Caenorhabditis* species were scanned for matches to known TF DNA-binding motifs (Fig. 2a). I observed that 153/210 TFs present in at least 5 species have a significantly enriched overlap across species of target orthologues with the TF motif (Wilcoxon test, FDR < 0.2), consistent with the notion of conserved TF regulation. For each potential TF-target pair, a conservation probability was calculated based on the random likelihood of identifying the observed pattern of motif conservation (taking into account both the number of species with significant motif matches and the strength of motif matches). Top motif matches for each TF in *C. elegans* were filtered by conservation probability (Supplementary Fig. 5a). TFs with few remaining targets after filtering (38/252 TFs) were removed. This led to a GRN which had a mean of 202.4 targets per TF (median 191.5). I observed that conserved motif-derived targets for those TFs that also had ChIP-seq data were 1.74 times more likely to feature a TF ChIP peak overlapping their promoter ($\chi^2$ test, 95% CI 1.64–1.84, P-value $2.91 \times 10^{-84}$), suggesting that they were indeed more likely to represent true TF targets.

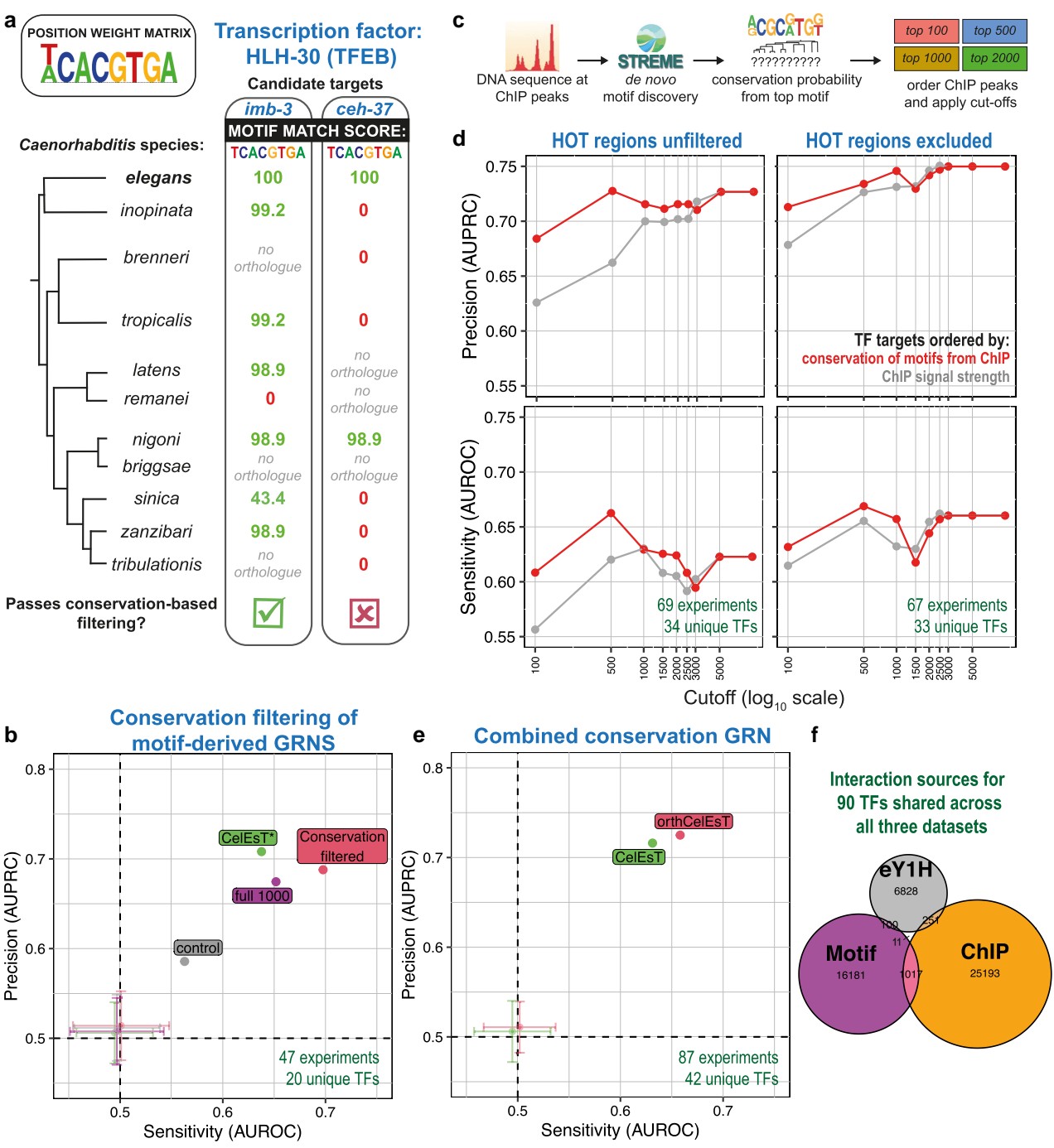

**Fig. 2.** Identification of cross-species conservation of TF binding motifs in orthologous target genes improves network performance with fewer TF targets. a) Example of two potential target genes (*imb-3* and *ceh-37*) with a perfect match to the HLH-30/TFEB motif in their promoters in *C. elegans*. Looking across 10 additional *Caenorhabditis* species, the HLH-30 motif is found in the *imb-3* promoter in 6/7 species with one-to-one orthologues, whereas the motif is present in the *ceh-37* promoter in only 1/7 species. b) Benchmarking performance for motif-based GRNs after conservation-based filtering ("Conservation filtered") vs the best-performing *elegans*-only motif-based GRN ("full 1000"), a control network with the same number of targets per TF selected from the best *elegans* motif matches ("control") and the *Cel*EsT network after subsetting to the same TFs ("CelEsT*"). Axes show performance (AUROC/AUPRC) in benchmarking pipeline. Transparent points show mean AUROC/AUPRC of 100 randomly shuffled networks; error bars show standard deviation. The number of experiments and unique TFs in the benchmarking set overlapping with TFs in the GRN is noted on the panel in green text. Results are shown for the best-performing multivariate model ("mlm") method; see also Supplementary Fig. 5b for performance with other statistical methods. c) Schematic shows application of conservation-based filtering to ChIP-seq data. After a de novo motif search from DNA sequences marked by ChIP-seq peaks, conservation probabilities are derived using the top resulting motif. Potential targets are then ordered by conservation probability, rather than ChIP peak signal strength as before, before applying cutoffs to compile GRNs. d) Line shows change in AUPRC (above) and AUROC (below) with increasing cutoff for maximum number of targets per TF for GRNs derived from ChIP-seq data with (right) or without (left) exclusion of target genes within HOT regions. Colors indicate network with targets ordered by conservation probability (upper line) or ChIP peak signal strength (lower line). Note, x-axis (cutoff) is on a log scale. Experiment/unique TF numbers in green text. See also Supplementary Fig. 5c. e) Benchmarking performance for orth*Cel*EsT, a combined GRN with conservation-based target prioritization applied to regulons of both the motif dataset and the ChIP-seq dataset. Results are shown for the best-performing multivariate model ("mlm") method; see also Supplementary Fig. 5d for performance with other methods. f) Euler plot shows TF-target interactions derived from each dataset for 90 TFs present in the orth*Cel*EsT network shared between all three datasets.

Despite featuring ~7.5× fewer targets on average, this GRN performed better than the much larger motif-derived GRN using the optimal cutoff of 1000 top *C. elegans* targets (Fig. 2b, Supplementary Fig. 5b). Importantly, it strongly outperformed a control GRN with a matched number of targets for each TF but consisting instead of the top motif matches in *C. elegans*. Thus, a cross-species conservation-based approach is very effective for filtering TF-target interactions to drastically reduce GRN size without loss of performance.

To test whether conservation-based target filtering could also be applied to ChIP data, de novo DNA-binding motifs were extracted from ChIP-seq data for 359 TFs (see Methods). Although conservation-based filtering of ChIP GRNs did not outperform unfiltered networks, it was observed that when TF targets were prioritized by conservation probability rather than by ChIP signal strength (Fig. 2c), performance was substantially improved at lower cutoffs, whether HOT regions were excluded or not (Fig. 2d). Indeed, for a ChIP-based GRN with HOT regions excluded, performance matched that of the full network at a regulon size cutoff of 1,000 targets (Fig. 2d; this cutoff affected 47 promiscuous TFs). Interestingly, GRNs compiled using de novo motifs derived from ChIP data performed markedly better than using the in vitro derived DNA-binding motif where available (119 TFs; Supplementary Fig. 5c).

A combined GRN was created with both motif- and ChIP-derived targets filtered by conservation as described above, together with the eY1H dataset. This GRN outperforms the *Cel*EsT network in the benchmarking pipeline using the "mlm" method (Fig. 2e, Supplementary Fig. 5d), but with an average of 57% fewer targets per TF and a more equal balance between interactions derived from different datasets (Fig. 2f). However, this comes at the loss of 18 TFs in the final network, for a total of 469. This combined conservation-based network is called orth*Cel*EsT (Supplementary Table 4).

Though improving network performance, both exclusion of ChIP peaks in HOT regions and conservation-based filtering of motif occurrences led to loss of TFs from the final network. This may affect some researchers with an interest in a particular TF which was excluded but for which data nonetheless exists. With conservation-based ChIP target ordering affording greatly improved performance at low cutoffs without filtering out HOT regions (Fig. 3d), it was possible to assemble a network with decent performance with no loss of TFs. Combining the top 500 conservation-prioritized ChIP targets together with the unfiltered top 1,000 motif-derived targets and the eY1H data produces a network with 506 TFs in total. The performance of this network, called max*Cel*EsT (Supplementary Table 5), compared favorably with *Cel*EsT using the "mlm" method (Supplementary Fig. 5e).

## *Cel*EsT recapitulates known TF biology in intensively studied conditions

To validate the performance of *Cel*EsT, TF activity estimation was performed on a variety of published RNA-seq datasets representing well-studied genetic, environmental, and physiological conditions. For each condition, independent studies were identified and analyzed for differential TF activity separately before a z-score was calculated for each TF to report the strength and consistency of differential activity across datasets (Fig. 3a).

First, datasets from mutants of the insulin/IGF1-like signaling (IIS) pathway were analyzed, focusing on the severe insulin receptor mutant *daf-2*(*e1370*) (Fig. 3b, Supplementary Fig. 6). Amalgamating 9 studies (Supplementary Fig. 7a; Supplementary Table 6), it was clear that activation (Fig. 3b), though not

upregulation (Supplementary Fig. 8a), of the FOXO orthologue DAF-16 was the major determinant of the transcriptional response to *daf-2* mutation, as is well established (Cahill *et al.* 2001; Lin *et al.* 2001). However, a consistent and strong enrichment was also observed for differentially-expressed targets of PHA-4 and depletion of the targets of NHR-80 (Fig. 3b). In 2 studies combining *daf-2* mutation with *daf-16* null mutations (Supplementary Fig. 7b) these changes are reversed (Supplementary Fig. 6b), suggesting that these TFs act downstream of DAF-16 activity. Although there is a significant DAF-16 ChIP peak in the promoters of both *pha-4* and *nhr-80*, neither gene is strongly regulated at the transcriptional level in *daf-2* mutants (Supplementary Fig. 8a). Similar reversed enrichments are found in data from mutation of the PTEN orthologue *daf-18* in a *daf-2* background (1 study, Supplementary Fig. 6c), which is known to suppress *daf-2* mutant phenotypic defects (Ogg and Ruvkun 1998), and in data from *daf-16* mutants in a wildtype background (3 studies, Supplementary Fig. 7c, Supplementary Fig. 6d). The latter result implies significant baseline DAF-16 activity despite normal levels of IIS signaling. Interestingly in this case, strong activity enrichments for NHR-25 and NHR-23 are seen that are not seen in a *daf-2* mutant background.

Data were examined from worms undergoing heat shock (Fig. 3c) from 6 RNA-seq studies (Supplementary Fig. 7d; Supplementary Table 6). As expected, the strongest and most consistent enrichment was for HSF-1 (heat shock factor 1; Fig. 3c). *hsf-1* mRNA barely changed despite strongly increased HSF-1 activity (Supplementary Fig. 8b). More surprising was the even more significant depletion of the targets of NHR-28, suggesting that this nuclear hormone receptor has a major role in the transcriptional response to heat shock. Although *nhr-28* mutants have been shown to have strongly impaired survival of heat shock (Joshi *et al.* 2016), this role has been overlooked until now. As *Cel*EsT is an unsigned network, this depletion would also be consistent with an increased activity of NHR-28 in repressing target genes; however, *nhr-28* itself appears to be downregulated at the transcriptional level upon heat shock (Supplementary Fig. 8b).

The transcriptional response to infection by pathogenic strains of the bacterium *P. aeruginosa* (11 studies; Supplementary Fig. 7e) was strongly dominated by a single TF, ZIP-2 (Fig. 3d). ZIP-2 has been previously been described as one key mediator of the early transcriptional response to *P. aeruginosa* infection (Estes *et al.* 2010) among others including PQM-1 (Rajan *et al.* 2019) and ATF-7 (Fletcher *et al.* 2019). While those TFs also display increased activity, the extent to which ZIP-2 determines the transcriptional response to infection suggests that it is of primary importance. ZIP-2 and other TFs that have significantly increased activity exhibited strongly increased expression (Supplementary Fig. 8c), demonstrating that, unlike in heat shock, the response to *P. aeruginosa* infection is largely orchestrated by regulation of TFs at the transcriptional level.

Lastly, differences in TF activity between males and hermaphrodites/pseudo-females were examined (3 studies; Supplementary Fig. 7f, Supplementary Table 6). The TF most strongly active in males relative to hermaphrodites was HLH-30 (Fig. 3e), despite no difference in expression (Supplementary Fig. 8d). HLH-30 nuclear exit in hermaphrodites is induced by mating (Shi *et al.* 2019), which raises the possibility that this difference is not intrinsic but is instead due to low activity of HLH-30 in hermaphrodites raised in the presence of males. In hermaphrodites, the E2F-like TFs EFL-1 and EFL-2 and their putative cofactor DPL-1 were more active (Fig. 3e), likely due to crucial roles in the hermaphrodite germline (Chi and Reinke 2009).

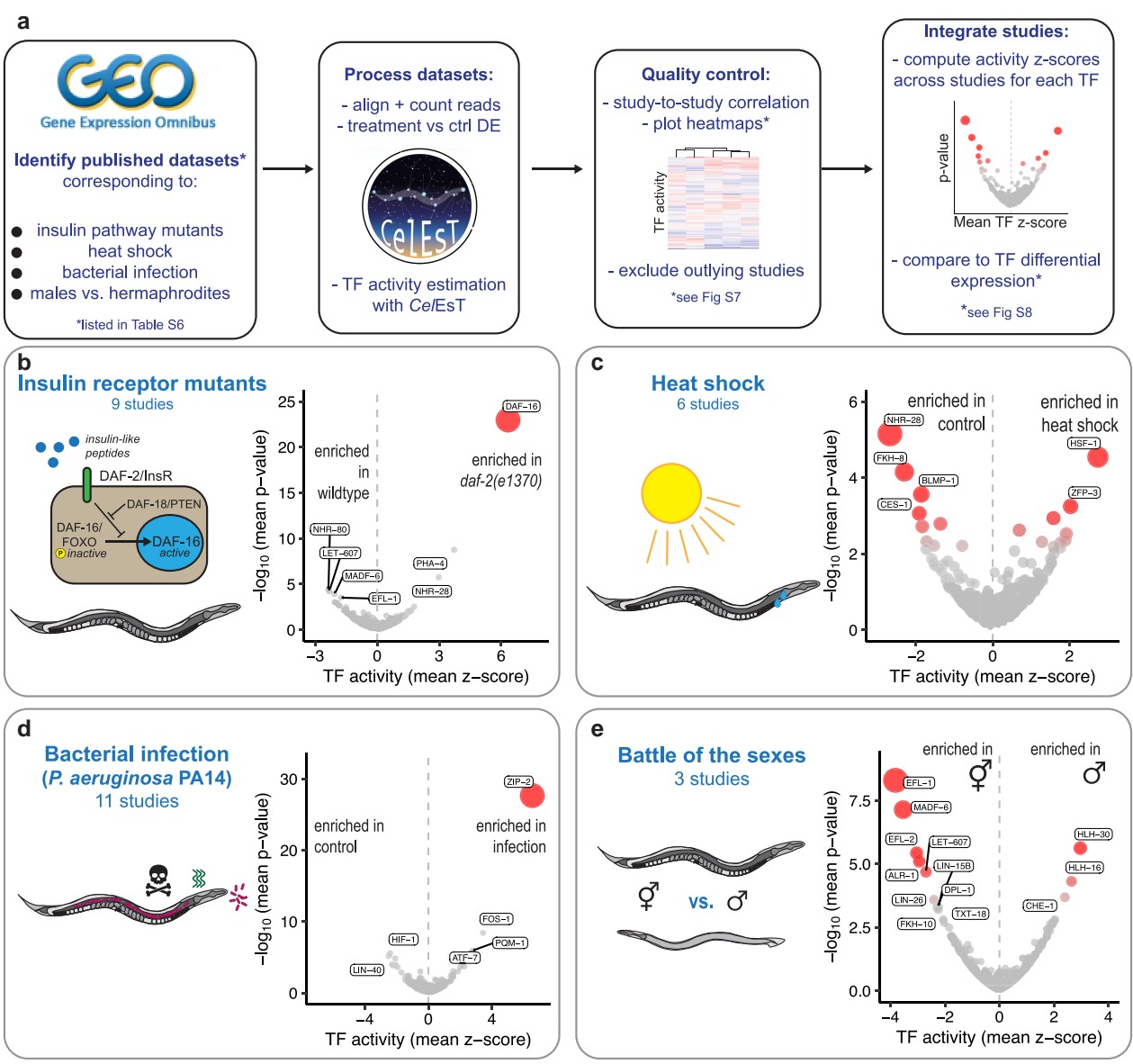

**Fig. 3.** *CelEsT* recapitulates known TF biology and generates new insights in commonly studied genetic, environmental, and physiological conditions. Volcano plots shows mean TF activity z-scores and geometric mean *P*-values across studies. The bubble size is proportional to the *P*-value. All analyses controlled for developmental age during DE analysis, with DE stats then analyzed using the multivariate linear model TF activity estimation method and the *CelEsT* network. a) Schematic of pipeline to integrate TF activity estimation analysis for multiple studies. b) Analysis of 9 studies (Supplementary Table 6) comparing a severe mutant allele of the insulin receptor orthologue *daf-2* to wildtype controls. See also Supplementary Fig. 6, Supplementary Fig. 7a, and Supplementary Fig. 8a. c) Analysis of 6 studies (Supplementary Table 6) comparing heat-shocked animals to untreated control animals. See also Supplementary Fig. 7d and Supplementary Fig. 8b. d) Analysis of 11 studies (Supplementary Table 6) comparing animals exposed to the pathogenic bacterium *P. aeruginosa* PA14 to untreated controls. See also Supplementary Fig. 7e and Supplementary Fig. 8c. e) Analysis of 3 studies (Supplementary Table 6) comparing the transcriptome of male animals to that of hermaphrodites. See also Supplementary Fig. 7f and Supplementary Fig. 8d.

In sum, these results both validate the use of *CelEsT* for TF activity estimation and highlight its potential to generate novel insights.

## Discussion

TF activity estimation from the expression of known target genes is a potent and increasingly popular method that allows inference of mechanistic insights from transcriptomic data. However, without prior knowledge of TF targets this method cannot be applied. Here, inspired by the human GRN DoRothEA (Garcia-Alonso *et al.* 2019), disparate large-scale experimental datasets previously generated by *C. elegans* researchers were integrated to provide a unified knowledge resource, the *CelEsT* GRN, thus making this powerful analysis available to the worm community. Importantly, *CelEsT* has been benchmarked against a set of TF perturbation RNA-seq experiments to ensure optimal processing and integration of distinct datasets.

*CelEsT* can be used with the *decoupleR/decoupler* package (Badia-i-Mompel *et al.* 2022) in either *R* or *Python* to infer TF activity estimates from gene expression data within minutes on a standard computer. I provide an annotated and customizable *R* script to users who wish to use *CelEsT* with specific parameter settings or alternative methods (Supplementary File 1). However, an *R* Shiny application is also provided for download at github.com/IBMB-MFP/CelEsT-app so that members of the community can

analyze their own data with minimal coding required. The *Cel*EsT app permits input of either existing DE statistics or raw sample gene-level read counts. In the latter case, the *Cel*EsT app can perform DE analysis and also allows the user to opt to correct for developmental age in order to eliminate any spurious differential gene expression derived from effects of their treatment on development (Bulteau and Francesconi 2022). Step-by-step guides to the use of both the *Cel*EsT app and the customizable *R* script can be found at Perez (2024a) and Perez (2024b) respectively.

Although multiple statistical methods for estimating TF activity were tested, it was shown that for these networks the multivariate linear model ("mlm") method consistently performed best. Importantly, this method considers multiple TFs in a single calculation, which allows for disentangling the distinct effects of multiple TFs even when their targets exhibit significant overlap. This method also happens to be the fastest and least computationally demanding of those tested. As such, it is recommended that users that employ the GRNs provided here use the "mlm" method of TF activity estimation for best performance.

The principal aim of this study was not to provide exhaustive lists of targets for each TF, but rather to find a core functional set of targets for each TF that acted as an accurate reporter for activity estimation. While the use of the *Cel*EsT GRN is recommended, the orth*Cel*EsT and max*Cel*EsT networks are also presented, which minimize TF-target interactions and TF loss at the cost of TF numbers and performance, respectively. The orth*Cel*EsT GRN, with ~40% of the interactions of the main *Cel*EsT network, represents a valuable resource for those seeking high-confidence TF targets in *C. elegans*. Meanwhile, the max*Cel*EsT GRN may be useful to researchers with an interest in a particular TF, which was ultimately excluded from the *Cel*EsT network.

Assigning targets to TFs based on putative matches to DNA-binding motifs is inherently likely to produce false positives (Garcia-Alonso *et al.* 2019). Here, it was shown that using conservation-based target filtering in combination with TF activity estimation benchmarking is a powerful method to winnow down potential motif-based targets to a highly informative set for each TF. This is likely effective precisely because it helps to eliminate false positives. Conservation of TF-binding motifs across species in the promoters of their targets is a strong indication that a given target does not represent a spurious sequence match but is a biologically relevant target that leads to selectable phenotypic impacts when TF regulation is lost (Tuğrul *et al.* 2015). Importantly, this method is highly applicable to a host of model organisms; in the age of comparative genomics it is common that multiple congeneric species of favorite models have available genome sequences (Hernández-Plaza *et al.* 2023).

It was also shown that a similar approach is applicable to ChIP-seq data to highlight biologically relevant targets based on conservation of de novo motifs. Targets marked by ChIP peaks are less likely than those with putative motif instances to be false positives, and strong filtering of ChIP-based targets according to de novo motif conservation did not improve network performance (not shown). However, motif conservation was a more effective metric than ChIP peak signal for target prioritization when applying stringent cutoffs to the number of targets per TF. Surprisingly, this method was effective despite using only one of several de novo motifs, which often did not correspond to in vitro determined motifs where known (see Methods). Unlike in vitro motifs from experiments using only the DNA-binding domain (Narasimhan *et al.* 2015), the de novo motifs reflect in vivo TF binding with a full-length protein, a full range of potential cofactors

and interaction partners and a native chromatin context. The top de novo motifs discovered may not represent the direct sequence preference of a TF's DNA-binding domain but likely also include motifs for relevant regulatory cofactors (Liu *et al.* 2018). Regardless, the higher performance observed using de novo motifs over known in vitro motifs where applicable shows that de novo motifs represent a potent biological signal which aids in identifying the most important targets of a TF.

One important limitation of *Cel*EsT is that it is an unsigned network that does not assign a mode of regulation (i.e. activator, repressor or bifunctional) to each TF. As such, greater activity of a repressor will lead to a depletion of its target mRNAs and thus to an apparently lower estimated activity. Further, TF activity estimation methods are known to be less effective for bifunctional TFs (Garcia-Alonso *et al.* 2019). While a subset of *C. elegans* TFs have an assigned mode-of-regulation in WormBase or UniProt (Supplementary Table 1), these annotations are far from complete and their inclusion somewhat impaired benchmarking performance (not shown). In any case, the tendency toward positive correlations between TF expression and activity (Supplementary Fig. 8) suggests that the dominant trend is toward positive TF regulation of targets.

Although comprehensively representing the bulk of *C. elegans* TF perturbation RNA-seq experiments conducted to date, another weakness is the limited size of the benchmarking set, both in terms of experiments and unique TFs represented. It is likely that with a larger benchmarking dataset and greater TF coverage, benchmarking performance would be less noisy and more likely to result in network parameters that provide optimal performance across all TFs. Expansion of a benchmarking dataset by future efforts toward a consortium or database providing systematic transcriptomic characterization of TF knock-out/knock-down animals would pay dividends for the *C. elegans* community.

In conclusion, the introduction of *Cel*EsT to enable TF activity estimation is an invaluable addition to the toolbox of the worm community. In addition, this study provides a roadmap for researchers who wish to effectively utilize existing knowledge resources for TF activity estimation in other model organisms. I hope that *Cel*EsT will not only enable more effective interpretation of future experiments but also allow a re-evaluation of past experiments to generate many novel insights, even from old data.

## Data availability

No original datasets were produced in this study. Availability of all datasets used is described in Methods and Supplementary Tables 2, 6, and 7. Supplementary Tables 3 and 5 are available on FigShare (https://doi.org/10.25386/genetics.27612672.v1). All analysis code is present on Github at github.com/IBMB-MFP/CelEsT-MS under a Creative Commons Zero 1.0 license.

Supplemental material available at GENETICS online.

## Acknowledgments

I thank Alexandra Avgustinova, Andre Faure, Mirko Francesconi, and Adam Klosin for critical comments on the manuscript. Thanks to Charlie Cotton for inspiring a suitably celestial logo, to my niece Celeste Perez for inspiring the name, to Jennifer Semple for use of her excellent worm illustrations and to Pau Badia-i-Mompel for always helpful, polite, and rapid responses to questions posted on the *decoupler* GitHub issues page. I thank Marian Walhout and one anonymous reviewer for their comments which helped to improve the manuscript. Supplementary Fig. 2d contains the Protein of the Month illustration of HIF-1

(credit: David S. Goodsell and the RCSB PDB) modified in agreement with the conditions of its CC-BY-4.0 license.

## Funding

This work was funded by a Ramon y Cajal fellowship (RYC2021-034496-I) awarded by Spain's Ministerio de Economía, Comercio y Empresa.

## Conflicts of interest

The author declares no conflict of interest.

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

*Editor: C. Kaplan*