## [Peer Review File · Genetics]

CeEsT: a unified gene regulatory network for estimating transcription factor activities in *C. elegans*

Marcos Francisco Perez

NOTE: The reviews and decision letters are unedited and appear as submitted by the reviewers.

In extremely rare instances and as determined by a Senior Editor or the EIC, portions of a review may be redacted. If a review is signed, the reviewer has agreed to no longer remain anonymous.

The review history appears in chronological order.

Review Timeline:

Submission Date:	2024-06-28
Editorial Decision:	2024-08-01
Resubmission Received:	2024-09-27
Accepted:	2024-11-02

August 1, 2024

GENETICS-2024-307234

CeEsT: a unified gene regulatory network for estimating transcription factor activities in *C. elegans*

Dear Dr. Perez:

Allow me to apologize for the delay in getting these reviews to you. We had some challenge finding reviewers, but I think they have given very useful reviews to help you improve the manuscript. Two experts in the field have reviewed your manuscript, and I have read it as well. The reviewers indicate that this application could be quite useful for the community but both reviewers note some drawbacks that need to be addressed before acceptance could be considered. While your manuscript is not currently acceptable for publication in GENETICS, we would welcome a substantially revised manuscript. Both reviewers have comments and concerns to be addressed in a revised manuscript. You can read their reviews at the end of this email. Most importantly, the manuscript is not viewed as accessible to the audience most likely to use the tool (see Reviewer 3 especially). Both reviewers wonder if analyses are sensitive to a subset of TFs and therefore some sensitivity analyses are warranted (Reviewer 3 suggests a "leave out" approach). Reviewer 1 is especially concerned about the RNAseq mapping strategy (see point 4). These comments need to be rigorously addressed and effort made to make the manuscript more clear and accessible to the audience that would likely use the analyses. We look forward to receiving your revised manuscript. Please let the editorial office know approximately how long you expect to need for revisions.

Upon resubmission, please include:

1. A clean version of your manuscript;
2. A marked version of your manuscript in which you highlight significant revisions carried out in response to the major points raised by the editor/reviewers (track changes is acceptable if preferred);
3. A detailed response to the editor's/reviewers' feedback and to the concerns listed above. Please reference line numbers in this response to aid the editor and reviewers.

Your paper will likely be sent back out for review.

Additionally, please ensure that your resubmission is formatted for GENETICS

<https://academic.oup.com/genetics/pages/general-instructions>

Follow this link to submit the revised manuscript: Link Not Available

Sincerely,

Craig Kaplan
Associate Editor
GENETICS

Approved by:
Karen Arndt
Senior Editor
GENETICS

Reviewer #1 (Comments for the Authors (Required)):

The author presents the integration of several gene regulatory networks (GRNs) in *C. elegans*, created using three types of protein-DNA interactions. Using expression data, the author estimated the active transcription factors (TFs) in the context of the corresponding target genes. Then, the author assessed the performance of various active estimators with expression data from TF perturbation assays. Additionally, the author demonstrated that integrated networks yield slightly better results. The author also highlighted that filtering target genes based on conservation across multiple species can improve TF activity estimation. Finally, through showcases, the author presented evidence that the TF activity discussed here has regulatory associations previously reported in the literature. Overall, this reviewer found the ideas, methods, and results presented here interesting and potentially impacting a broad audience. However, this reviewer also thinks several significant comments must be addressed, particularly those referring to clarity and reasoning behind some of the analysis presented and its interpretation.

1. This reviewer finds several sections quite confusing. For instances:

- a. Lines: 101-109. It is presented as an introduction, but actually is describing the methods used to build the different GRNs.
 - b. lines 111: "Three large-scale experimental datasets report TF-target interactions in *C. elegans*". What is the purpose of this title? A more analysis-aimed description could help here. For instance: "Establishment of GRNs based on several data types."
 - c. Lines 217-218: The out says "While the 'mlm' method was still the best performing individual method, unlike the ChIP-based GRNs the motif-based GRN also performed well with both 'ulm' and 'wsum' (Fig S2B-C)." This claim is confusing because Figure S2C presents "consensus" as having the best performance, yet this isn't mentioned anywhere else.
2. Lines 252-254: "This combined and weighted GRN performs well, with an AUPRC of 0.722 and an AUROC of 0.641 (Fig 1E), a performance comparable to popular human GRNs 254 like DoRothEA (MÜLLER-DOTT et al. 2023)." This note could apply to any of the performances analyzed by the author: Although these values are the highest within the GRNs and methods, they are not necessarily high, considering that 0.5 is the value we can get from a random prediction. This reviewer wonders if this could be related to the genes expressed in the corresponding perturbation assays. In other words, is the network reduced to only genes in the perturbation assay? Perhaps the author could enrich the performance discussion by considering these details.
3. Lines 316-317: "Interestingly, GRNs compiled using de novo motifs derived from ChIP data performed markedly better than using the known DNA-binding motif where available (119 TFs; Fig S5C)." This quote, along with every Venn diagram (or Euler diagram as referred to by the author), is extremely confusing (Figs 1G, 2F, S3D, and S5E). How can the author explain such a low overlap in integrations predicted from motifs and ChIP-seq, especially when the motifs and ChIP come from the same TF? Also, how can it be explained that the individual performances (motif vs. ChIP) are not significantly lower than when they are both integrated (Fig S3)? I wonder if perhaps the performance metrics are being biased or driven for only a few TFs with high-quality data that oversahed the other TFs. It could be very valuable to the author by evaluating AUPCR and AUROC values for individual TFs to avoid potential overestimations of performance.
4. Lines 853-841: The author indicates that all the perturbed and control samples were mapped using bowtie2, which definitely is not the best option to perform mapping of RNA-seq libraries given its incapacity to map spliced reads. STAR or HISAT2 are more suitable tools for mapping RNA-seq data. HISAT2 in particular, is a follow up from the same lab that reported Bowtie2 (PMID: 31375807). This could represent a serious flaw of this study given that all activity estimations, as well as differential expression analyses, are clearly inclined to "see" lower expression values in every single gen with an intron on it. This reviewer thinks that issues should be addressed in order for this work to get published in this or any other journal.
5. At the beginning of the results section, the author states that "TF activity" can be estimated using expression data and predefined GRNs. The author then introduces the methods (ulm, mlm, and wsum) but does not explain what these methods are or how they define activity or how to interpret "TF activity." This reviewer assumed that the author refers to the beta values of the regression, correct? If so, it is unclear why the activities are not marked as activation or repression, as mentioned in the discussion (lines 529-537).
6. Supplementary file S1(Customizable_R_script.txt), could greatly benefit the community and improve the reproducibility of the reported results if it were presented as a GitHub repository and organized by figures or at least by analysis type.

Minor comments:

- All supplemental materials need descriptions and names on the files themselves.
- There is a typo in Fig S2a: "homoytpic"
- Lines 331: Perhaps the author refers to Fig 2D instead of "Fig 3D"?
- Line 814: "To produce a set of TF perturbation RNA-seq experiments" perhaps the author meant "to collect/gather/compile" are better options.

Reviewer #3 (Comments for the Authors (Required)):

This paper aims to combine available ChIP and eY1H data together with TF binding motifs and benchmarks true target prediction with RNA-seq data to predict gene regulatory networks (GRNs). Doing this is long overdue and, therefore, the paper is a welcome contribution to the field of gene regulation in *C. elegans*. Integration the evolutionary conservation of TF binding motifs is also a good addition.

Detailed comments:

- The paper is hard to follow in many places. It would be much recommended if the author focuses on how a *C. elegans* geneticist would use the tool and provide a step-by-step guide. If time and resources permit, a searchable website would be helpful.
- The author appears to have missed a paper in which we have directly compared TF binding by ChIP or Y1H to promoter activity. This should be discussed and included. PMID: 26430702.
- I strongly feel the RNA-seq data used to benchmark should not be called 'comprehensive' since it is (as the author states) very

limited in size.

- The first paragraph of the Results section is hard to read and its purpose is not clear.
 - A total of 833 TFs is mentioned. Where does this number come from? We have also extensively curated predicted TFs, which the author could consider including (e.g., in the FuxmanBass paper and in PMID: 23791784).
 - I think it would be very useful to delve deeper into the lack of overlap among the three datasets used, why is this and what could cause it?
 - I think Figure 1 should include the diagram of only overlapping TFs (as in Fig. 1G, now in Supplement).
 - The author should delve deeper into different types of TFs - which families do better with which assay (in predictive performance)?
 - How much are the predictions driven by which TFs? Redoing the analyses by systematically removing the best performing TF and seeing how the predictions then perform would be helpful to see how generally useful the GRNs are.
 - It is unclear to me why certain cutoffs were chosen (e.g., 15 promoters in the eY1H data) - could this cause overfitting?
 - The part on the detailed RNA-seq studies was hard to read. Precisely what was done? A cartoon would be helpful. Also, were these RNA-seq datasets included in the benchmarking? If so, that is not good and the author should redo the benchmarking without each dataset and then perform the testing?
- I hope these comments help the author to improve the paper.

Marian Walhout

Associate Editor Comments:

Response to reviewers for:
**Perez 2024; CelEsT: a unified gene regulatory network for estimating
transcription factor activities in *C. elegans***

Reviewers comments are marked in bold purple text

Author's response is marked in black

"Excerpts from manuscript quoted and indented, 10 pt Times New Roman and italics"

Excerpts retain markup from manuscript revisions:

"Text in blue has been added to the original manuscript"

Text in green has been moved to a new location in the manuscript

Numbers in purple reflect numbers which have changed in the latest analysis

*Calls to figures or supplementary tables are **highlighted in yellow**"*

[Line numbers indicated below in square brackets]

Reviewer #1

The author presents the integration of several gene regulatory networks (GRNs) in *C. elegans*, created using three types of protein-DNA interactions. Using expression data, the author estimated the active transcription factors (TFs) in the context of the corresponding target genes. Then, the author assessed the performance of various active estimators with expression data from TF perturbation assays. Additionally, the author demonstrated that integrated networks yield slightly better results. The author also highlighted that filtering target genes based on conservation across multiple species can improve TF activity estimation. Finally, through showcases, the author presented evidence that the TF activity discussed here has regulatory associations previously reported in the literature. Overall, this reviewer found the ideas, methods, and results presented here interesting and potentially impacting a broad audience. However, this reviewer also thinks several significant comments must be addressed, particularly those referring to clarity and reasoning behind some of the analysis presented and its interpretation.

1. This reviewer finds several sections quite confusing. For instances:
a. Lines: 101-109. It is presented as an introduction, but actually is describing the methods used to build the different GRNs.

The reviewer is correct that this material was not well placed in the original manuscript. It has now been relocated further down within the first section of the Results, as below. Green text reflects the specified portion of text in its new position:

"In order to judge the performance of the GRNs resulting from the integration of the three data resources described above, I used the benchmarking pipeline from the decoupler package (Fig 1a) (BADIA-I-MOMPEL et al. 2022). This pipeline assesses the ability of TF activity estimations to correctly identify perturbed TFs from expression of their target genes in a benchmarking transcriptomic dataset. This allows for a quantitative performance comparison both for different GRNs and for different statistical methods of estimating TF activity, using the classifier metrics AUROC and AUPRC. Although AUROC/AUPRC are related and correlated metrics, they can display opposite trends (e.g. Fig S1c) and so GRNs that maximised both AUROC and AUPRC were prioritised where possible. Of the methods which gauge TF activity by fitting a linear model to target gene expression, the univariate linear model ('ulm') method considers TFs separately, whereas the multivariate linear model ('mlm') method fits a single model to all TFs. The latter can thereby theoretically disentangle the separate effects of distinct TFs with overlapping targets. The weighted sum method ('wsun') involves summing scores for all targets of a given TF. The decoupler package can also provide a consensus score which integrates the scores from disparate methods." [lines 123 - 137]

b. lines 111: "Three large-scale experimental datasets report TF-target interactions in *C. elegans*". What is the purpose of this title? A more analysis-aimed description could help here. For instance: "Establishment of GRNs based on several data types."

This title has been changed to:

"Establishment of GRNs based on large-scale datasets of direct TF-DNA interactions"
[line 103]

c. Lines 217-218: The out says "While the 'mlm' method was still the best performing individual method, unlike the CHIP-based GRNs the motif-based GRN also performed well with both 'ulm' and 'wsum' (Fig S2B-C)." This claim is confusing because Figure S2C presents "consensus" as having the best performance, yet this isn't mentioned anywhere else.

The reviewer is correct that consensus appears to have the best performance. However, the consensus score does not reflect a specific method but rather integrates the scores from the other methods. In order to avoid confusion I have added some words of explanation:

"While the 'mlm' method was still the best performing individual method, the motif-based GRNs also performed well with both 'ulm' and 'wsum' (Fig S2b-c), leading to the best performance from the 'consensus' score which integrates results from multiple methods."

[lines 218 - 220]

2. Lines 252-254: "This combined and weighted GRN performs well, with an AUPRC of 0.722 and an AUROC of 0.641 (Fig 1E), a performance comparable to popular human GRNs 254 like DoRothEA (MÜLLER-DOTT et al. 2023)." This note could apply to any of the performances analyzed by the author: Although these values are the highest within the GRNs and methods, they are not necessarily high, considering that 0.5 is the value we can get from a random prediction. This reviewer wonders if this could be related to the genes expressed in the corresponding perturbation assays. In other words, is the network reduced to only genes in the perturbation assay? Perhaps the author could enrich the performance discussion by considering these details.

Although the AUROC/AUPRC values reported may not be high by the standards of e.g. a diagnostic classifier, they compare favourably to established methods of TF activity estimation which are widely and productively used by researchers, such as the popular human DoRothEA network and promise to provide useful biological insights from transcriptomic experiments, as demonstrated in the last section of the paper.

The GRN is never reduced during the benchmarking assays. In the benchmarking set, the GRN is not altered to limit it only to the TFs in the perturbation assay. Likewise, the network is not limited to genes identified as expressed in the perturbation assays. In the first instance, the targets considered for the GRN are genes which tend to be consistently detected across whole-animal RNA-seq samples in the CeNDR dataset, which is the largest bulk *C. elegans* RNA-seq dataset (ZHANG et al. 2022).

Values for genes unexpressed or otherwise excluded from the DE analysis in the perturbation assays for benchmarking are set to 0. I have experimented with excluding these genes – leaving them as 'NA' – in calculating the benchmarking performance. However to do so increased the computational demand several orders of magnitude and made no perceptible difference to the results.

3.Lines 316-317: "Interestingly, GRNs compiled using de novo motifs derived from CHIP data performed markedly better than using the known DNA-binding

motif where available (119 TFs; Fig S5C)." This quote, along with every Venn diagram (or Euler diagram as referred to by the author), is extremely confusing (Figs 1G, 2F, S3D, and S5E).

That conserved *de novo* ChIP-derived motifs performed better than conserved *in vitro*-derived motifs from PBM/SELEX experiments was indeed a surprising result, which thus merited inclusion in the manuscript and some place in the discussion.

Importantly, the 'known' motifs from CisBP are not necessarily (and exclusively) 'true': they are derived from *in vitro* experiments conducted on the TF's DNA binding domain (DBD) alone with no proteome or chromatin context. Additionally, the Hughes lab which conducted the majority of the PBM experiments (WEIRAUCH *et al.* 2014; NARASIMHAN *et al.* 2015) chose a single 'representative' motif for each DBD from multiple motifs extracted from their data by different algorithms.

The *de novo* motifs derived in the manuscript from ChIP data are from *in vivo* TF binding with a full-length protein, a full range of potential cofactors and interaction partners and a native chromatin context. It is therefore plausible that these motifs may sometimes be more biologically relevant than the *in vitro*-derived DBD motif – and indeed these results suggest that.

I have expanded the discussion a little to reflect this:

"However motif conservation was a more effective metric than ChIP peak signal for target prioritisation when applying stringent cut-offs to the number of targets per TF. Surprisingly this method was effective despite using only one of several de novo motifs, which often did not correspond to in vitro- determined motifs where known (see Methods). Unlike in vitro motifs from experiments using only the DNA-binding domain (NARASIMHAN et al. 2015), the de novo motifs reflect in vivo TF binding with a full-length protein, a full range of potential cofactors and interaction partners and a native chromatin context. The top de novo motifs discovered may not represent the direct sequence preference of a TF but likely also include motifs for relevant regulatory co-factors (LIU et al. 2018). Regardless, the higher performance observed using de novo motifs over known motifs where applicable shows that de novo motifs encode a potent biological signal which aids in identifying the most important targets of a TF."

[lines 541 - 552]

How can the author explain such a low overlap in integrations predicted from motifs and ChIP-seqs, especially when the motifs and ChIP come from the same TF?

In response to both this question and a suggestion from reviewer #3, I have added an exploration of the different characteristics of targets derived from motif and ChIP data which sheds some light on the biases of each strategy and goes some way to explaining the limited overlap:

"ChIP-derived interactions were biased towards target genes that were located in active chromatin domains (Fig S3e; (EVANS et al. 2016)) and that were expressed either broadly across tissues (Fig S3f) or in tissues that contribute disproportionately to whole-animal chromatin (up to 32n polyploid intestinal cells (MCGHEE 2007) and the multinucleate gonad/germline; Fig S3g). Motif-derived targets showed no bias for either chromatin context or expression across tissues (Fig S3e-g)."

[lines 263 - 268]

e) Fraction of target genes derived from ChIP-seq or promoter motifs which are found in stable 'active' or 'regulated' chromatin domains, as defined in (EVANS *et al.* 2016). p-values from chi-squared test.

f) Violin plot of Gini coefficient for gene expression across tissues for TF targets derived from ChIP-seq or promoter motifs. p-values are from Kruskal Wallis/Dunn's post-hoc tests. Gini coefficient for each gene was calculated using tissue-specific gene expression for L2 larvae reported by (CAO *et al.* 2017). A Gini coefficient of 1 indicates highly unequal expression, whereas a low Gini coefficient indicates equal expression across tissues.

g) Enrichments for tissue-specific genes (reported by (CAO *et al.* 2017)) among target genes derived from ChIP-seq or promoter motifs. The number of tissue-specific genes for each tissue is indicated in parentheses after the tissue type on the x axis. Error bars show 95% confidence intervals. Bubble size indicates p-value (Fisher's exact test).

NS, not significant.

It is important to note that this lack of overlap is very similar to what has been reported by others in the past. In the 2019 paper that presented the widely-employed (>600 citations) human GRN DoRothEA (GARCIA-ALONSO *et al.* 2019) which inspired this *C. elegans*-specific effort, the lack of overlap between targets derived from different data sources was highly reminiscent of what I report in this manuscript, as seen in their Fig 1D. Although they employ a different graphical format to report this result, I have replotted the data from their figure in the style I employ in this manuscript – as a three-group Euler diagram (here excluding their coexpression-derived TF-target interactions, but the general result is robust with any three groupings) - to make the similarity more explicit.

Also, how can it be explained that the individual performances (motif vs. ChIP) are not significantly lower than when they are both integrated (Fig S3)?

In fact, individual performances of the motif and ChIP networks are indeed significantly lower than when integrated, judged with a benchmark set of common TFs and using the multivariate linear model (mlm) method which emerges as the method of choice in this manuscript. To make this clearer I have moved this out of the supplementary and into Fig 1, as Fig 1e (below).

This boost to performance upon network combination may not appear very impressive, but we can suppose that each network carries with it a proportion of false positive or negligibly important interactions which act to limit its predictive accuracy. Upon combination of the networks these weak interactions are still included and still act to limit the effectiveness of the combined network, such that its performance, while improved, is not dramatically better than that of the single networks.

I wonder if perhaps the performance metrics are being biased or driven for only a few TFs with high-quality data that overshadow the other TFs. I could be very valuable to the author by evaluating AUPRC and AUROC values for individual TFs to avoid potential overestimations of performance.

In response to this suggestion and similar feedback from Reviewer #3 I have added a new supplementary figure (Fig S4) where I provide benchmarking performance for individual TFs. It must be borne in mind that these figures will not be accurate, both due to the very limited size of the benchmarking set for any individual TF and for the few ranked elements in benchmarking classifiers using single TFs. Nonetheless it

shows, perhaps unsurprisingly, a range of performances, with some TFs no better than random to some with a consistent perfect classification. Most TFs show intermediate performance. I also performed an analysis where I remove ~25% (10/42) of the TFs from the benchmarking set at a time and repeat the benchmarking performance metrics for 1000 iterations; while performance is sometimes improved and sometimes worsened, the GRN always performs substantially better than random.

“While most major TF families displayed good performance, the zinc finger NHR and C2H2 families did not perform better than random (Fig S4a-b). I note however that the benchmarking set covers <2% of TFs in these families. Performances for individual TFs, while approximate due to tiny benchmarking sets, varied substantially (Fig S4c-d). Notably, performance of some TFs depended strongly on whether developmental age correction was applied. While the putative developmental clock gene *BLMP-1* (*MEEUSE et al. 2020*) performed markedly worse with age correction, which may remove some signal from *BLMP-1*-driven transcriptomic changes, *ZTF-11* performed much better. The performance of the *CelEsT* GRN was robust to removal of ~25% of TFs from the benchmarking set (Fig S4e-f) and thus was not largely driven largely by a small number of high-performing TFs.”

[lines 270 - 279]

Fig S4. *CelEsT* performance is not driven by a small number of high-quality TFs

a, b) Benchmarking performance for TFs from major families with (a) or without (b) correction for inferred developmental age. The point size reflects the number of benchmarking experiments; the size of

the interior ring reflects the number of unique TFs. Faded points and error bars reflect mean/SD of AUROC/AUPRC for each family for 100 random shuffled networks.

c, d) Benchmarking performance for individual TFs with (c) or without (d) correction for inferred developmental age. Point colour reflects TF family; point size reflects number of benchmarking experiments. The best/worst TFs are labelled individually. Points labelled in c are also labelled in d and vice versa.

e, f) Benchmarking performance with (e) or without (f) correction for inferred developmental age for the Ce/EsT network (red dot) and upon removal of 10 TFs at a time from the benchmarking set (~25% of the total). Each grey dot represents one of 1000 trials removing a subset of TFs.

Legend below show colours for families, as well as benchmarking experiments, unique TFs and total TFs for each.

4. Lines 853-841: The author indicates that all the perturbed and control samples were mapped using bowtie2, which definitely is not the best option to perform mapping of RNA-seq libraries given its incapacity to map spliced reads. STAR or HISAT2 are more suitable tools for mapping RNA-seq data. HISAT2 in particular, is a follow up from the same lab that reported Bowtie2 (PMID: 31375807). This could represent a serious flaw of this study given that all activity estimations, as well as differential expression analyses, are clearly inclined to "see" lower expression values in every single gen with an intron on it. This reviewer thinks that issues should be addressed in order for this work to get published in this or any other journal.

I thank the reviewer for flagging this important point. All analyses have been repeated using alignment conducted with HISAT2 and all figures have been regenerated.

I note that this has made only a small difference to the overall results. This may be because in this manuscript the TF activity estimations have all been performed on differential expression statistics, which result from comparison across samples of particular genes quantified in the same way for each sample (ie. with or without splicing awareness for all factors). Therefore use of bowtie2 in this case, while inappropriate, did not strongly influence the outcome of the analyses. This is evidenced by the high correlation (median ~0.98) of the test statistics for benchmarking experiments between the pipelines with each

aligner (see boxplot, in which each point represents one study in the benchmarking set).

One subtle change which did have some downstream ramifications was to change the best-performing cutoff in the motif GRNs to the top 1000 targets (previously top 1500 had performed marginally better). This led to a change in the Ce/EsT GRN that produced some differences in later results, for example in Fig 3, although the principal results were not altered. Similarly, another small change was to shift the best performing FDR cutoff for the motif conservation analysis from 0.5 to 0.8, somewhat

increasing the average targets in the orthology-filtered motif-based GRNs from ~160 to ~200.

5. At the beginning of the results section, the author states that "TF activity" can be estimated using expression data and predefined GRNs. The author then introduces the methods (ulm, mlm, and wsum) but does not explain what these methods are or how they define activity or how to interpret "TF activity." This reviewer assumed that the author refers to the beta values of the regression, correct? If so, it is unclear why the activities are not marked as activation or repression, as mentioned in the discussion (lines 529-537).

The reviewer is correct that the TF activity estimate is equivalent to the beta values of the regressions, for the two regression-based methods (ulm and mlm).

The point in the discussion refers to the GRN being unsigned (i.e. activators and repressors are not distinguished), not to the methods failing to mark particular TF's regulons as enriched or depleted. The targets of a repressor that is more active in the treatment condition will appear significantly depleted; that repressor will therefore have a strongly negative 'TF activity' (= strongly negative regression beta-value). However this is not distinguishable (without other information) from strongly reduced activity of an activating TF. Therefore the sign of the activity/beta-value cannot reliably indicate increased or reduced TF activity without knowing the mode of action of the particular TF under consideration.

To point interested readers to more information about the methods employed, I have added a line to direct them to a more detailed overview in the paper describing the *decoupler* package:

"Of the statistical methods which gauge TF activity by fitting a linear model to target gene expression, the univariate linear model ('ulm') method considers TFs separately, whereas the multivariate linear model ('mlm') method fits a single model to all TFs. The latter can thereby theoretically disentangle the separate effects of distinct TFs with overlapping targets. The weighted sum method ('wsum') involves summing scores for all targets of a given TF. The decoupler package can also provide a consensus score which integrates the scores from disparate methods. For more detail on specific methods, see Table S1 of (BADIA-I-MOMPEL et al. 2022)."

[lines 131 - 138]

6. Supplementary file S1(Customizable_R_script.txt), could greatly benefit the community and improve the reproducibility of the reported results if it were presented as a GitHub repository and organized by figures or at least by analysis type.

All scripts used to reproduce all analyses are present in a Github repository at github.com/IBMB-MFP/CeEsT-MS, arranged hierarchically and numbered according to dependence on the output of the execution of previous scripts. Code chunks to reproduce certain figure panels are annotated throughout these scripts. I have now added the customisable script from Supplementary File 1 to this Github repository.

The purpose of the customisable script is to provide researchers with only the code to perform the basic analysis with a minimum of interpretation and personalisation. I also provide the *CeEsT* Shiny app to make the analysis available in an almost entirely

coding-free manner to members of the community. Further, in response to a suggestion from reviewer #3 I provide detailed step-by-step guides on protocols.io that can be found here and here.

Minor comments:

- All supplemental materials need descriptions and names on the files themselves.

I have inserted names and descriptions into the files themselves, which are now in Excel format to allow for names and descriptions in the first sheet.

- There is a typo in Fig S2a: "homoytpic"

I thank the reviewer for their sharp eye; this typo has now been corrected

- Lines 331: Perhaps the author refers to Fig 2D instead of "Fig 3D"?

Again, thanks to the reviewer for their close attention to the manuscript. The reviewer is correct and this has now been rectified.

- Line 814: "To produce a set of TF perturbation RNA-seq experiments" perhaps the author meant "to collect/gather/compile" are better options.

I have amended the text to use 'compile' instead of 'produce' as suggested.

Reviewer #3

This paper aims to combine available ChIP and eY1H data together with TF binding motifs and benchmarks true target prediction with RNA-seq data to predict gene regulatory networks (GRNs). Doing this is long overdue and, therefore, the paper is a welcome contribution to the field of gene regulation in *C. elegans*. Integration the evolutionary conservation of TF binding motifs is also a good addition.

Detailed comments:

- The paper is hard to follow in many places. It would be much recommended if the author focuses on how a *C. elegans* geneticist would use the tool and provide a step-by-step guide. If time and resources permit, a searchable website would be helpful.

In order to provide *C. elegans* geneticists with the tools they need to perform TF activity estimation analysis, I have added step-by-step detailed guide on protocols.io can be found here and here, which respectively tell users how to download and use the Ce/EsT Shiny app and to use the script provided for customised analysis. I have now highlighted these step-by-step protocol in the introduction and discussion so that those who wish to use it can find what they need without having to wade through the details in the paper.

My initial intention was for the Ce/EsT app to be available for use on a web platform – however the momentary peak of memory usage at the point of fitting the mlm model is such that the cost of hosting it is prohibitive. For this reason it is instead available for download from Github for use on local machines.

As for searchable resources for researchers looking to identify targets, the supplementary table S3, S4 and S5 provide this. However to make them more useful I have expanded the annotations on these tables to provide the TF and target identifiers also as WormBase Gene IDs and as gene names, as below:

source	target	weight	with motif	in ChIP	in eY1H	TF WBgeneID	TF name	target WBgeneID	target name
B0304.1	2L52.1	0.666666667	TRUE	TRUE	FALSE	WBGene00001948	hlh-1	WBGene00007063	2L52.1
C07H6.7	2L52.1	0.333333333	NA	TRUE	FALSE	WBGene00003024	lin-39	WBGene00007063	2L52.1
C17H12.9	2L52.1	0.333333333	TRUE	FALSE	FALSE	WBGene00015934	ceh-48	WBGene00007063	2L52.1
C27A12.5	2L52.1	0.333333333	TRUE	FALSE	FALSE	WBGene00000429	ceh-2	WBGene00007063	2L52.1

- The author appears to have missed a paper in which we have directly compared TF binding by ChIP or Y1H to promoter activity. This should be discussed and included. PMID: 26430702.

I have incorporated this reference into the introduction:

“Indeed, previous analysis of the transcriptional effects of TF perturbations in the C. elegans intestine found that physical interaction between TFs and promoters often underlies direct control of target gene expression (MACNEIL et al. 2015).”

[lines 47 - 50]

- I strongly feel the RNA-seq data used to benchmark should not be called 'comprehensive' since it is (as the author states) very limited in size.

Although the benchmarking set is limited in size, the use of the word ‘comprehensive’ was intended to convey that this set includes almost all of the *C. elegans* TF perturbation RNA-seq experiments that I could identify in the literature and in the GEO (with a few deliberate exclusions; as mentioned in the Methods section, in order to not bias the set of experiments too strongly in favour of particular intensively-studied TFs such as DAF-16, I applied a maximum of 4 experiments per unique TF in the benchmarking set). However in accordance with the reviewer’s objection this description has been removed from the text, except in the following passage in the Discussion where the sense of the word’s usage is more clear:

“Although comprehensively representing the bulk of C. elegans TF perturbation RNA-seq experiments conducted to date, another weakness is the limited size of the benchmarking set, both in terms of experiments and unique TFs represented. It is likely that with a larger benchmarking dataset and greater TF coverage, benchmarking performance would be less noisy and more likely to result in network parameters that provide optimal performance across all TFs.”

[lines 564 - 571]

- The first paragraph of the Results section is hard to read and its purpose is not clear.

The reviewer is correct that this material was not well placed in the original manuscript. It has now been relocated further down within the first section of the Results, as below. Green text reflects the specified portion of text in its new position:

“In order to judge the performance of the GRNs resulting from the integration of the three data resources described above, I used the benchmarking pipeline from the decoupler package (Fig 1a) (BADIA-I-MOMPEL et al. 2022). This pipeline assesses the ability of TF activity estimations to correctly identify perturbed TFs from expression of their target genes in a benchmarking transcriptomic dataset. This allows for a quantitative performance comparison both for different GRNs and for different statistical methods of estimating TF activity, using the classifier metrics AUROC and AUPRC. Although AUROC/AUPRC are related and correlated metrics, they can display opposite trends (e.g. Fig S1d) and so GRNs that maximised both AUROC and AUPRC were prioritised where possible. Of the statistical methods which gauge TF activity by fitting a linear model to target gene expression, the univariate linear model (‘ulm’) method considers TFs separately, whereas the multivariate linear model (‘mlm’) method fits a single model to all TFs. The latter can thereby theoretically disentangle the separate effects of distinct TFs with overlapping targets. The weighted sum method (‘wsum’) involves summing scores for all targets of a given TF. The decoupler package can also provide a consensus score which integrates the scores from disparate methods.”

[lines 123 - 137]

- A total of 833 TFs is mentioned. Where does this number come from? We have also extensively curated predicted TFs, which the author could consider including (e.g., in the FuxmanBass paper and in PMID: 23791784).

I thank the reviewer for pointing out that the source of this number is not made explicit. This is important given that there are several figures which are quoted in the literature (FUXMAN BASS *et al.* 2016; MA *et al.* 2021; KUDRON *et al.* 2024).

This number quoted in this manuscript comes from a curated list in the latest paper from the modERN consortium (KUDRON *et al.* 2024). While during development of the project I was working from the reviewer's curated list wTF3.0 published in FUXMAN BASS *et al.* 2016, upon publication of the KUDRON *et al.* 2024 preprint I switched to using this list as it represents the most recent curated list of *C. elegans* TFs.

I have now made this clear in the Methods section:

"Identities and total number of C. elegans TFs"

Despite multiple published estimates (FUXMAN BASS et al. 2016; MA et al. 2021; KUDRON et al. 2024), the quoted number of C. elegans TFs at 833 is derived from the curated list in the latest manuscript from the modERN consortium (KUDRON et al. 2024). TF family annotations were derived from the wTF3 resource (FUXMAN BASS et al. 2016)." [lines 873 - 878]

- I think it would be very useful to delve deeper into the lack of overlap among the three datasets used, why is this and what could cause it?

I thank the reviewer for prompting me to investigate further, which has helped to shed some light on the biases inherent in the ChIP-seq and motif datasets which might explain their divergence. I have added several new figure panels in Fig S3 and a new paragraph to explain these differences:

"ChIP-derived interactions were biased towards target genes that were located in active chromatin domains (Fig S3e; (EVANS et al. 2016)) and that were expressed either broadly across tissues (Fig S3f) or in tissues that contribute disproportionately to whole-animal chromatin (up to 32n polyploid intestinal cells (MCGHEE 2007) and the multinucleate gonad/germline; Fig S3g). Motif-derived targets showed no bias for either chromatin context or expression across tissues (Fig S3e-g)." [lines 263 - 268]

e) Fraction of target genes derived from ChIP-seq or promoter motifs which are found in stable 'active' or 'regulated' chromatin domains, as defined in (EVANS *et al.* 2016). p-values from chi-squared test.

f) Violin plot of Gini coefficient for gene expression across tissues for TF targets derived from ChIP-seq or promoter motifs. p-values are from Kruskal Wallis/Dunn's post-hoc tests. Gini coefficient for each gene was calculated using tissue-specific gene expression for L2 larvae reported by (CAO *et al.* 2017). A Gini coefficient of 1 indicates highly unequal expression, whereas a low Gini coefficient indicates equal expression across tissues.

g) Enrichments for tissue-specific genes (reported by (CAO et al. 2017)) among target genes derived from ChIP-seq or promoter motifs. The number of tissue-specific genes for each tissue is indicated in parentheses after the tissue type on the x axis. Error bars show 95% confidence intervals. Bubble size indicates p-value (Fisher's exact test).

NS, not significant.

- I think Figure 1 should include the diagram of only overlapping TFs (as in Fig. 1G, now in Supplement).

Fig 1h (previously Fig 1G) has been replaced with the figure including only overlapping TFs that was previously in the supplement.

- The author should delve deeper into different types of TFs - which families do better with which assay (in predictive performance)?

- How much are the predictions driven by which TFs? Redoing the analyses by systematically removing the best performing TF and seeing how the predictions then perform would be helpful to see how generally useful the GRNs are.

In response to these two points, I have produced a new supplementary figure with accompanying paragraph investigating performance by TF family, by individual TFs, and testing the robustness of the network performance to TF removal by removing ~25% of TFs at a time from the benchmarking set.

“While most major TF families displayed good performance, the zinc finger NHR and C2H2 families did not perform better than random (Fig S4a-b). I note however that the benchmarking set covers <2% of TFs in these families. Performances for individual TFs, while approximate due to tiny benchmarking sets, varied substantially (Fig S4c-d). Notably, performance of some TFs depended strongly on whether developmental age correction was applied. While the putative developmental clock gene BLMP-1 (MEEUSE et al. 2020) performed markedly worse with age correction, which may remove some signal from BLMP-1-driven transcriptomic changes, ZTF-11 performed much better. The performance of the CelEst GRN was robust to removal of ~25% of TFs from the benchmarking set (Fig S4e-f) and thus was not largely driven largely by a small number of high-performing TFs.”

[lines 270 - 279]

Fig S4. CelEsT performance is not driven by a small number of high-quality TFs

a, b Benchmarking performance for TFs from major families with (a) or without (b) correction for inferred developmental age. The point size reflects the number of benchmarking experiments; the size of the interior ring reflects the number of unique TFs. Faded points and error bars reflect mean/SD of AUROC/AUPRC for each family for 100 random shuffled networks.

c, d Benchmarking performance for individual TFs with (c) or without (d) correction for inferred developmental age. Point colour reflects TF family; point size reflects number of benchmarking experiments. The best/worst TFs are labelled individually. Points labelled in c are also labelled in d and vice versa.

e, f Benchmarking performance with (e) or without (f) correction for inferred developmental age for the CelEsT network (red dot) and upon removal of 10 TFs at a time from the benchmarking set (~25% of the total). Each grey dot represents one of 1000 trials removing a subset of TFs.

Legend below show colours for families, as well as benchmarking experiments, unique TFs and total TFs for each.

- It is unclear to me why certain cutoffs were chosen (e.g., 15 promoters in the eY1H data) - could this cause overfitting?

The minimum cutoff of 15 targets per TF was chosen arbitrarily. Rather than causing overfitting, the aim was to avoid misleading TF activity estimates based on linear model fitting of only a few data points. I note that the FUXMAN BASS *et al.* 2016 paper applies a similar cutoff of 10 interactions before linear model fitting for TF mode-of-action prediction (Fig 3B), presumably with a similar rationale. Nonetheless it so happens that the minimum target cutoff of 15 gives better performance than using 5, 10 or 20:

I note that although this cutoff leads to a large number of excluded TFs from the eY1H data in the analysis of a GRN based on this dataset alone (with 160 TFs passing the cut-off based on only eY1H interactions), the eY1H data contributes some interactions for many more TFs in the final network (275 TFs), as this arbitrary minimum cutoff of 15 targets is applied to the total of interactions from all data types after they are combined.

- The part on the detailed RNA-seq studies was hard to read. Precisely what was done? A cartoon would be helpful. Also, were these RNA-seq datasets included in the benchmarking? If so, that is not good and the author should redo the benchmarking without each dataset and then perform the testing?

To make interpretation of this section clearer, as suggested I have added a cartoon as Fig 3a to make more clear what analysis underlies the remaining plots in Figure 3.

The benchmarking set includes only direct TF perturbation experiments, and in Fig 3 the treatments all represent either environmental/physiological conditions or mutations of non-TF genes (in the case of *daf-2*). As such, there is no overlap at all between the benchmarking dataset and the studies shown in Fig 3.

One minor exception is one of the two *daf-2(e1370);daf-16(mu86)* mutant studies shown in Fig S6b. *daf-16* mutation is the only direct TF perturbation in this collection of figures; even so, only one of the 5 *daf-16* mutant samples (between panels b and d) used here is included in the benchmarking set. Despite this small overlap I feel its inclusion in this figure dissecting TF activity in IIS mutants is strongly merited, as the main point being made in the text is not the reduced DAF-16 activity evident in these samples but rather the DAF-16 dependence of other TFs with altered activity in *daf-2* mutants:

“However a consistent and strong enrichment was also observed for differentially-expressed targets of PHA-4 and depletion of the targets of NHR-80 (Fig 3b). In 2 studies combining daf-2 mutation with daf-16 null mutations (Fig S7b) these changes are reversed (Fig S6b), suggesting that these TFs act downstream of DAF-16 activity.”

[lines 409 - 413]

I also feel that to exclude this high-quality experiment from the benchmarking set based on its inclusion in Fig S6b would be to throw the baby out with the bathwater. Nonetheless, to acknowledge the reviewer's valid point I have added a note in the figure legend for Fig S6b to make clear that one of these two studies does indeed overlap with the benchmarking set:

“(b) 2 studies (see Fig S7b and Table S6) comparing daf-2(e1370); daf-16(mu86) to daf-2(e1370). Note that one of these two studies was included in the benchmarking set used to evaluate GRN performance.”

[lines 801 - 803]

I hope these comments help the author to improve the paper.

Marian Walhout

Thank you, Marian.

REFERENCES FOR RESPONSE:

- Badia-i-Mompel, P., J. Vélez Santiago, J. Braunger, C. Geiss, D. Dimitrov *et al.*, 2022 decoupleR: ensemble of computational methods to infer biological activities from omics data. *Bioinformatics Advances* 2: vbac016.
- Cao, J., J. S. Packer, V. Ramani, D. A. Cusanovich, C. Huynh *et al.*, 2017 Comprehensive single-cell transcriptional profiling of a multicellular organism. *Science* 357: 661-667.
- Evans, K. J., N. Huang, P. Stempor, M. A. Chesney, T. A. Down *et al.*, 2016 Stable *Caenorhabditis elegans* chromatin domains separate broadly expressed and developmentally regulated genes. *Proceedings of the National Academy of Sciences* 113: E7020-E7029.
- Fuxman Bass, J. I., C. Pons, L. Kozlowski, J. S. Reece-Hoyes, S. Shrestha *et al.*, 2016 A gene-centered *C. elegans* protein–DNA interaction network provides a framework for functional predictions. *Molecular Systems Biology* 12: 884.
- Garcia-Alonso, L., C. H. Holland, M. M. Ibrahim, D. Turei and J. Saez-Rodriguez, 2019 Benchmark and integration of resources for the estimation of human transcription factor activities. *Genome Research* 29: 1363-1375.
- Kudron, M., L. Gewirtzman, A. Victorsen, B. C. Lear, J. Gao *et al.*, 2024 Binding profiles for 954 *Drosophila* and *C. elegans* transcription factors reveal tissue specific regulatory relationships. *bioRxiv*: 2024.2001. 2018.576242.
- Liu, B., J. Yang, Y. Li, A. McDermaid and Q. Ma, 2018 An algorithmic perspective of de novo cis-regulatory motif finding based on ChIP-seq data. *Briefings in Bioinformatics* 19: 1069-1081.
- Ma, X., Z. Zhao, L. Xiao, W. Xu, Y. Kou *et al.*, 2021 A 4D single-cell protein atlas of transcription factors delineates spatiotemporal patterning during embryogenesis. *Nature Methods* 18: 893-902.
- MacNeil, L. T., C. Pons, H. E. Arda, G. E. Giese, C. L. Myers *et al.*, 2015 Transcription factor activity mapping of a tissue-specific in vivo gene regulatory network. *Cell Systems* 1: 152-162.
- McGhee, J. D., 2007 The *C. elegans* intestine. *WormBook: The Online Review of C. elegans Biology* [Internet].
- Meeuse, M. W., Y. P. Hauser, L. J. Morales Moya, G. J. Hendriks, J. Eglinger *et al.*, 2020 Developmental function and state transitions of a gene expression oscillator in *Caenorhabditis elegans*. *Molecular Systems Biology* 16: e9498.
- Narasimhan, K., S. A. Lambert, A. W. Yang, J. Riddell, S. Mnaimneh *et al.*, 2015 Mapping and analysis of *Caenorhabditis elegans* transcription factor sequence specificities. *eLife* 4: e06967.
- Weirauch, M. T., A. Yang, M. Albu, A. G. Cote, A. Montenegro-Montero *et al.*, 2014 Determination and inference of eukaryotic transcription factor sequence specificity. *Cell* 158: 1431-1443.
- Zhang, G., N. M. Roberto, D. Lee, S. R. Hahnel and E. C. Andersen, 2022 The impact of species-wide gene expression variation on *Caenorhabditis elegans* complex traits. *Nature Communications* 13: 3462.

October 21, 2024

RE: GENETICS-2024-307499

Dr. Marcos Francisco Perez
Institut de Biologia Molecular de Barcelona
Department of Cells and Tissues
C/ Baldiri Reixac, 4-8
Torre R, 3era Planta
Barcelona, N/A 08028
Spain

Dear Dr. Perez:

Congratulations, your manuscript entitled "CelEsT: a unified gene regulatory network for estimating transcription factor activities in *C. elegans*" is accepted for publication in GENETICS! Many thanks for submitting your research to the journal.

The reviewers had a few suggestions for improving the manuscript that you may want to consider. Reviewer 1 did note to me that the paper is still a bit dense and could be difficult to read. To maximize accessibility, I would recommend sharing the manuscript with a few trusting colleagues or additionally members of the field and soliciting comments on making it easier to read. You can view their comments at the bottom of this email.

To Proceed to Publication:

1. Format your article according to GENETICS style: <https://academic.oup.com/genetics/pages/general-instructions>

2. Ensure that you comply with data and community resource citation guidelines:

<https://academic.oup.com/genetics/pages/general-instructions#Data-Policy>

3. Upload your final files at <https://genetics.msubmit.net>

4. Add oupsupport@scipris.com and genetics.oup@novatechset.com (or the domains @scipris.com and @novatechset.com) to your email program's "safe senders" list. You will be contacted by both at various points during the production process.

Notes:

- Your currently-accepted manuscript (unedited, as submitted, reviewed, and accepted) will be published at GENETICS and deposited into PubMed as an Advance Access article. Notify sourcefiles@thegsajournals.org before signing your license if you do not wish to publish your article via Advance Access.

- We invite you to submit an original color figure related to your paper for consideration as cover art. Please email your submission to the editorial office or upload it with your final files. You can submit a small-sized image for evaluation, and if selected, the final image must be a TIFF file 2513px wide by 3263px high (8.375 by 10.875 inches; resolution of 600ppi). Please avoid graphs and small type.

- After files are sent to Oxford University Press we use SciPris to manage article licensing and payment. If you do not have a SciPris account, you will receive an email from no-reply@scipris.com to sign up to use Oxford University Press' author portal. After logging in, follow the online instructions to sign your license and arrange any payment due.

If you have any questions or encounter any problems while uploading your accepted manuscript files, please email the editorial office at sourcefiles@thegsajournals.org.

Sincerely,

Craig Kaplan
Associate Editor
GENETICS

Approved by:
Karen Arndt

Review comments (if applicable):

Reviewer #1 :

This reviewer appreciates the improvements and clarifications made by the author. However, there are a few minor comments that should be addressed.

Minor comments/clarification

- Line 42: what does the author mean with "transcriptional responses to insults"?
- Line 72: what does the author mean with "environmental insults"?
- Lines 135-127: "This pipeline assesses the ability of TF activity estimations to correctly identify perturbed TFs from expression of their target genes in a benchmarking transcriptomic dataset." This phrase is confusing; it sounds like the pipeline predicts target genes, but then it seems to select or prioritize TFs for an already defined set of target genes. Need clarification.
- Lines 279: "largely" is written twice.

Reviewer #3 :

The author has addressed all comments. I am still a bit concerned about thresholding and overfitting, but do think this can provide a useful resource for the community.